# Large Language Models are Efficient Learners of Noise-Robust Speech Recognition

**Yuchen Hu**[1,†*]   **Chen Chen**[1,*]   **Chao-Han Huck Yang**[2,3,†]
**Ruizhe Li**[4]   **Chao Zhang**[5]   **Pin-Yu Chen**[6]   **Eng Siong Chng**[1]
[1]Nanyang Technological University   [2]Georgia Institute of Technology   [3]NVIDIA Research
[4]University of Aberdeen   [5]Tsinghua University   [6]IBM Research

## Abstract

Recent advances in large language models (LLMs) have promoted generative error correction (GER) for automatic speech recognition (ASR), which leverages the rich linguistic knowledge and powerful reasoning ability of LLMs to improve recognition results. The latest work proposes a GER benchmark with "HyPoradise" dataset to learn the mapping from ASR N-best hypotheses to ground-truth transcription by efficient LLM finetuning, which shows great effectiveness but lacks specificity on noise-robust ASR. In this work, we extend the benchmark to noisy conditions and investigate *if we can teach LLMs to perform denoising for GER just like what robust ASR do*, where one solution is introducing noise information as a conditioner into LLM. However, directly incorporating noise embeddings from audio encoder could harm the LLM tuning due to cross-modality gap. To this end, we propose to extract a language-space noise embedding from the N-best list to represent the noise conditions of source speech, which can promote the denoising process in GER. Furthermore, in order to enhance its representation ability of audio noise, we design a knowledge distillation (KD) approach via mutual information estimation to distill the real noise information in audio embeddings to our language embedding. Experiments on various latest LLMs demonstrate our approach achieves a new breakthrough with up to 53.9% correction improvement in terms of word error rate while with limited training data. Analysis shows that our language-space noise embedding can well represent the noise conditions of source speech, under which off-the-shelf LLMs show strong ability of *language-space denoising*[1].

## 1 Introduction

Recent advances in large language models (LLMs) have attracted a surge of research interest due to their representation power of language generation (OpenAI, 2022; 2023; Touvron et al., 2023a), which achieve a wide range of success on natural language processing (NLP) tasks (Brown et al., 2020; Wei et al., 2022; Ouyang et al., 2022). Powered by LLMs, latest works (Chen et al., 2023b; Yang et al., 2023a) propose a generative error correction (GER) framework[2] for automatic speech recognition (ASR), along with a "HyPoradise" dataset[3] that contains abundant pairs of ASR N-best hypotheses and ground-truth transcription. It has shown great performance in learning the mapping from hypotheses to transcription by parameter-efficient LLM finetuning (Hu et al., 2021), which significantly outperforms typical LM rescoring methods (Mikolov et al., 2010). However, their study lacks specificity on noisy ASR scenarios, which are the most common in real world (Li et al., 2015).

In this work, we extend the GER benchmark to noisy conditions, as well as propose a Robust HyPoradise (RobustHP) dataset with 113K hypotheses-transcription pairs from various ASR corpus in common noisy scenarios. Similar to the original benchmark, we also observe error correction improvement of LLM finetuning on noisy ASR, but the performance gain in most noisy conditions is still limited (see Table 1). It indicates that LLMs-based GER is still prone to source audio noise

---

*Equal contribution. †Corresponding authors: `yuchen005@e.ntu.edu.sg`, `hucky@nvidia.com`

[1]This work is open sourced at: `https://github.com/YUCHEN005/RobustGER`
[2]`https://github.com/Hypotheses-Paradise/Hypo2Trans`
[3]`https://huggingface.co/datasets/PeacefulData/Robust-HyPoradise`

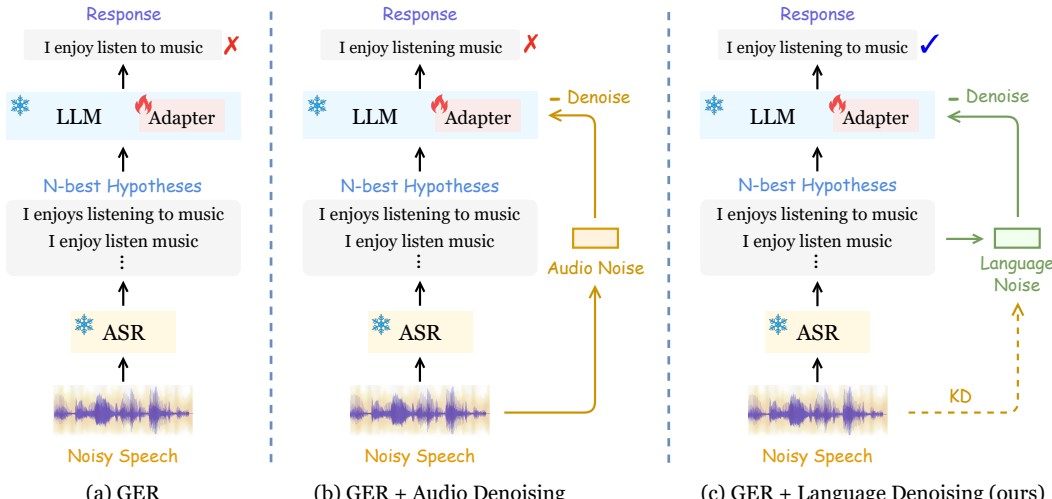

Figure 1: Overview of (a) GER (Chen et al., 2023b; Yang et al., 2023a), (b) GER with audio-space denoising (Zhang et al., 2023b) (see details in §B.1), (c) GER with language-space denoising.

(see our case study in Table 5). Luckily, we draw inspiration from the noise-robust ASR community. Their key idea is to map noisy speech features to clean space (i.e., denoise) before recognition (Li et al., 2014), where speech enhancement denoising (Pandey et al., 2021) is one of the most popular approaches. Therefore, we raise a research question for our case: *Can we teach LLMs to denoise the N-best hypotheses for GER, just like what robust ASR and speech enhancement do?*

Inspired by recent works on LLM adaptation (Wu et al., 2023a; Fathullah et al., 2023; Gao et al., 2023), a general solution here is to incorporate audio noise information as a conditioner into LLM finetuning to make it noise-aware, which is also similar to the popular conditional diffusion model (Dhariwal & Nichol, 2021). However, latest works find that directly introducing other modalities (*e.g.*, audio, visual) into LLM finetuning could harm its stability and performance due to cross-modality gap (Zhang et al., 2023b; Li et al., 2023b). Our examination in Table 1 also indicates this limitation.

To this end, we propose to extract a noise embedding in *language space* to represent the noise conditions of source speech, by measuring the diversity of N-best hypotheses list from ASR decoding. The insight behind is that, the worse noisy conditions (more challenging noise type or lower SNR), the higher uncertainty of ASR beam search decoding, and thus results in more diverse N-best hypotheses, which has been illustrated in Table 15 and Fig 6. Extracted from the language space of hypotheses instead of audio space, our noise embedding can be well incorporated into LLM tuning to improve GER, which can be viewed as a novel *language-space denoising* process. Furthermore, in order to enhance its representation ability of audio noise, we design a knowledge distillation (KD) approach via mutual information estimation (Belghazi et al., 2018) to distill the real noise information in audio embeddings to our extracted language embedding. As a result, it presents stronger noise representativeness (see Fig. 4(b)) and enhances the denoising performance. Various latest LLMs (*e.g.*, LLaMA-2 (Touvron et al., 2023b), LLaMA (Touvron et al., 2023a) and Falcon (Penedo et al., 2023)) are utilized to verify the effectiveness of our approach, and the comprehensive experimental results demonstrate that our model improves the GER performance with up to 53.9% word error rate (WER) reduction on RobustHP test sets while with limited training data.

Our contribution can be summarized as follows:

- We extend the latest ASR generative error correction benchmark to noise-robust ASR, where a Robust HyPoradise (RobustHP) dataset with 113K hypotheses-transcription pairs is collected from various ASR corpus in common noisy conditions.

- We propose RobustGER, a noise-aware generative error correction approach based on LLMs to map N-best hypotheses to true transcription, where an extracted language-space noise embedding with audio distillation is utilized to teach LLMs to perform denoising.

- Experiments on various latest LLMs show the proposed approach achieves a new break-through on RobustHP with up to 53.9% GER improvement in terms of word error rate

(WER). Analysis verifies the effectiveness of our proposed language-space embedding to represent audio noise, under which LLMs show strong ability of *language-space denoising*.

## 2    RELATED WORK

**Large Language Models and Parameter-efficient Finetuning.** There is recently a surge of research interests in Transformer-based LLMs, such as ChatGPT (OpenAI, 2022), GPT-4 (OpenAI, 2023) and LLaMA (Touvron et al., 2023a). Benefiting from giant model size and abundant training data, LLMs can understand the linguistic structures and semantic meanings behind text, which shows remarkable performance on a wide range of NLP tasks (Brown et al., 2020; Wei et al., 2022; Ouyang et al., 2022). To adapt LLMs to downstream tasks, many recent works investigate parameter-efficient LLM finetuning (Hu et al., 2021) considering its huge model size. In order to further exploit the potential of LLMs on multimodal tasks, more recent works investigate to incorporate other modalities into LLM tuning (Wu et al., 2023a; Fathullah et al., 2023; Li et al., 2023a; Chen et al., 2023c; Zhang et al., 2023a;b; Gao et al., 2023; Wang et al., 2023; Radhakrishnan et al., 2023). However, the latest works find that directly introducing other modalities into LLMs could harm the finetuning stability and performance due to the heterogeneous cross-modality gap (Zhang et al., 2023b; Li et al., 2023b). Therefore, this work proposes to extract a language embedding from the N-best list to represent audio noise, which works well in teaching LLMs to perform denoising.

**LM Rescoring and ASR Generative Error Correction.** LM rescoring has been widely used in ASR decoding to improve the linguistic acceptability of recognition results, which achieves stable gains of ASR performance (Arisoy et al., 2015; Shin et al., 2019; Mikolov et al., 2010; Yang et al., 2021; Yu et al., 2023). Typically, an external LM is deployed to rescore the N-best hypotheses list from ASR beam search decoding to rerank out the 1-best candidature. Furthermore, to make full use of all candidatures, recent works use the entire N-best list for error correction (Leng et al., 2021; Ma et al., 2023; Hu et al., 2020; 2023a; Guo et al., 2019; Hu et al., 2022a; Chen et al., 2023a), which outperforms rescoring methods. Powered by LLMs, the latest works propose generative error correction (GER) benchmark (Chen et al., 2023b) to directly predict the ground-truth transcription from ASR N-best hypotheses. To enable the learning of hypotheses-to-transcription mapping, they also propose a HyPoradise dataset with 316K hypotheses-transcription pairs. This work extends the GER benchmark to the most common noisy ASR scenarios with a new Robust HyPoradise dataset.

**Noise-robust ASR.** Neural ASR has achieved human-level performance in recent years but its noise-robustness in the real world remains a challenge (Krishna et al., 2019). Recent noise-robust ASR methods make some progress by mapping noisy speech features to clean space (i.e., denoise) before recognition (Li et al., 2014; Pandey et al., 2021). For instance, speech enhancement serves as a denoising front-end (Fu et al., 2019; Hu et al., 2023b) to improve speech quality for ASR (Hu et al., 2022b; 2023d;c), domain adversarial training aims to learn noise-invariant speech features (Prasad et al., 2021), and the recently popular ASR foundation model proposes to use web-scale data and various preprocessing steps for denoising (Radford et al., 2023). Inspired by them, this work investigates to teach LLMs to denoise the N-best hypotheses in language space for GER.

## 3    BENCHMARK AND DATASET

### 3.1    GENERATIVE ERROR CORRECTION BENCHMARK

We extend the original generative error correction benchmark (Chen et al., 2023b) to noise-robust ASR. Given an input noisy speech $X_n$, the pre-trained ASR model first transcribes it into $N$-best hypotheses $\mathcal{Y}_N = \{Y_1, Y_2, \cdots, Y_N\}$ by beam search decoding. The goal of GER is to learn a hypotheses-to-transcription (H2T) mapping $\mathcal{M}_{\text{H2T}}$ that predicts the transcription $Y$ based on $N$-best hypotheses list $\mathcal{Y}_N$:

$$Y = \mathcal{M}_{\text{H2T}}(\mathcal{Y}_N), \tag{1}$$

Given the ground-truth transcription $Y^*$, we can finetune the LLM to learn $\mathcal{M}_{\text{H2T}}$ in an auto-regressive manner, where the cross-entropy loss $\mathcal{L}_{\text{H2T}}$ is formulated as:

$$\mathcal{L}_{\text{H2T}} = \sum_{t=1}^{T} -\log \mathcal{P}_\theta(y_t^* | y_{t-1}^*, \cdots, y_1^*, \mathcal{Y}_N), \tag{2}$$

where $y_t^*$ is the $t$-th token of $Y^*$, and $\theta$ denotes the learnable parameters in LLM (i.e., adapter).

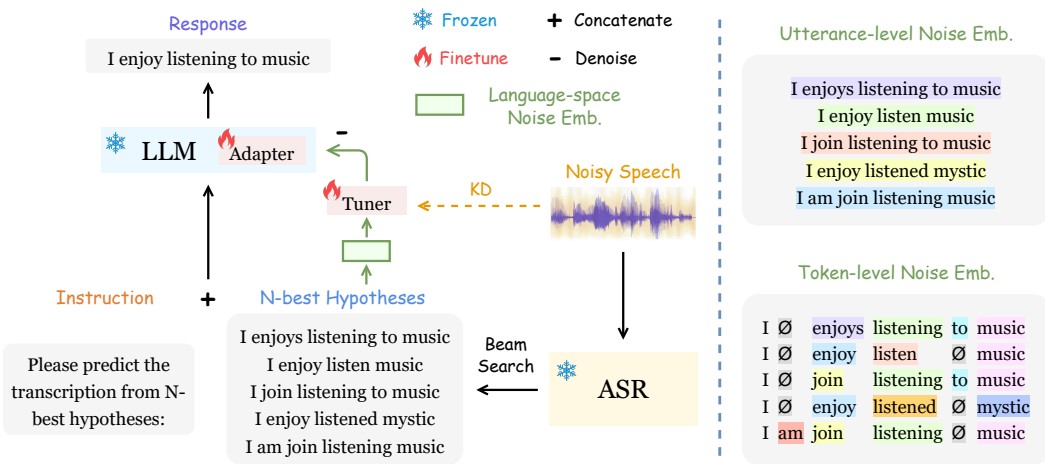

Figure 2: **Left:** The RobustGER framework that leverages efficient LLM finetuning to learn mapping from ASR N-best hypotheses to ground-truth transcription, where we propose a language-space noise embedding with audio distillation to denoise GER process. **Right:** The extraction of language-space noise embedding from N-best hypotheses by measuring its diversity, where we calculate the utterance- and token-level embedding differences between each pair of hypotheses in the N-best list. The details of embedding extraction are illustrated in §4.2 and Eq. (4)-(6).

## 3.2   ROBUST HYPORADISE DATASET

Correspondingly, we develop a Robust HyPoradise dataset by collecting hypotheses-transcription (HT) pairs from common noisy ASR corpus, including CHiME-4 (Vincent et al., 2016), VoiceBank-DEMAND (Valentini-Botinhao et al., 2016), NOIZEUS (Hu & Loizou, 2006), LibriSpeech-FreeSound (Prasad et al., 2021) and RATS (Graff et al., 2014), with details provided in §A. We employ Whisper Large-V2 (Radford et al., 2023), the state-of-the-art ASR foundation model to transcribe the noisy speech into N-best hypotheses (N is set to 5). As a result, we collect 113K HT pairs in total from various noise domains, and the dataset statistics are presented in Table 6.

## 4   METHOD

In this section, we present our noise-aware generative error correction (RobustGER) approach. We first describe the overall framework (§4.1), and then we introduce the extraction of language-space noise embedding from N-best hypotheses (§4.2), followed by audio noise distillation (§4.3) at last.

## 4.1   OVERALL FRAMEWORK

The left part of Fig. 2 presents the overall framework of RobustGER. First, the noisy speech $X_n$ is sent into a pre-trained ASR model to generate N-best hypotheses $\mathcal{Y}_N = \{Y_1, Y_2, \cdots, Y_N\}$, where $N = 5$. Following that, we propose to extract a language-space noise embedding $E_{\text{LN}}$ from the N-best list $\mathcal{Y}_N$ to represent the noise conditions of source speech $X_n$. As depicted in the right part of Fig. 2, such noise embedding measures the diversity of N-best hypotheses on both utterance and token levels, which perceives the noise information in input speech.

Furthermore, to enhance its noise representation ability, we design a KD approach to distill the real noise information in source speech $X_n$ to the extracted language-space noise embedding $E_{\text{LN}}$. Specifically, we employ the audio embedding $\mathcal{E}_{\text{ASR}}(X_n)$ from ASR encoder for distillation.

Finally, we add an instruction onto the N-best hypotheses and sent them into LLM to predict the true transcription (i.e., GER), with the language embedding incorporated for denoising. Specifically, we add a minus sign before the noise embedding $E_{\text{LN}}$ to indicate "**de**noise". Such minus embedding is then sent to teach LLM to do language-space denoising. Therefore, Eq.(1) should be re-written as:

$$Y = \mathcal{M}_{\text{H2T}}(\mathcal{Y}_N; -E_{\text{LN}}),   \tag{3}$$

The $\mathcal{M}_{\text{H2T}}$ denotes H2T mapping by efficient LLM finetuning, where we follow the adapter tuning from previous works (Zhang et al., 2023b; Yang et al., 2023b). We also borrow their idea of input-level prompting to incorporate our language noise embedding into LLM tuning, and the details are presented in §B.1. Similar to Eq.(2), we follow the original GER benchmark for optimization.

## 4.2 LANGUAGE-SPACE NOISE EMBEDDING

As directly incorporating audio-space noise embedding into LLM finetuning could harm its stability and performance (Zhang et al., 2023b; Gao et al., 2023), we propose an alternative to extract language-space noise embedding from N-best hypotheses to represent the noise conditions of source speech. The key idea is to perceive the audio noise from the diversity of N-best hypotheses, i.e., the worse noisy conditions (more challenging noise type or lower SNR), the higher uncertainty of ASR beam search decoding, and thus results in more diverse N-best hypotheses (see Table 15 and Fig 6).

As illustrated in the right part of Fig. 2, we extract the noise embedding on both utterance and token levels to capture rich diversity information: 1) *Utterance-level*: examine the diversity inside N-best list in terms of the entire utterance's semantic meaning, which indicates the affect of audio noise on the global semantics of hypotheses; 2) *Token-level*: examine the distribution of N-best hypothesis in terms of all the tokens inside, which is similar to edit distance and thus directly corresponds to the WER metric. These two embeddings are finally combined to form the resulted noise embedding, i.e., $E_{\text{LN}} = [E_{\text{LN}}^{utt}; E_{\text{LN}}^{tok}]$. Specifically, we employ sentence-BERT (SBERT) (Reimers & Gurevych, 2019) to obtain the embeddings from raw text, which contains rich language-space semantic information.

### 4.2.1 UTTERANCE-LEVEL NOISE EMBEDDING

Given N-best hypotheses $\mathcal{Y}_N = \{Y_1, Y_2, \cdots, Y_N\}$, we first obtain their sentence embeddings by SBERT encoder $\mathcal{E}_{\text{sbert}}$ and then calculate their diversity as:

$$E_{\text{LN}}^{utt} = \text{Concat}\{[\mathcal{E}_{\text{sbert}}(Y_i) - \mathcal{E}_{\text{sbert}}(Y_j)]_{i,j=1,i>j}^N\} \in \mathbb{R}^{\frac{N \cdot (N-1)}{2} \times D_{\text{sbert}}}, \quad (4)$$

where $D_{\text{sbert}}$ denotes the embedding size of SBERT extractor. In short, it concatenates all the sentence embedding differences $\mathcal{E}_{\text{sbert}}(Y_i) - \mathcal{E}_{\text{sbert}}(Y_j)$ where $i > j$, resulting in an utterance-level noise embedding $E_{\text{LN}}^{utt} \in \mathbb{R}^{N \cdot (N-1)/2 \times D_{\text{sbert}}}$. The key idea is, $Y_i$ ranks lower than $Y_j$ in the N-best hypotheses list, which thus presents lower confidence and worse transcription quality, i.e., more *language noise*. Therefore, Eq.(4) serves as a measurement of the audio noise in language space. The worse noisy speech would lead to larger ASR decoding uncertainty and thus more diverse N-best hypotheses, so that Eq.(4) can capture larger diversity embedding.

### 4.2.2 TOKEN-LEVEL NOISE EMBEDDING

Apart from utterance-level embedding, we also propose to extract token-level noise embedding that directly corresponds to the WER metric of ASR task. As shown in the bottom-right part of Fig. 2, similar to the calculation of edit distance, we first forced-align the N-best hypotheses to the same length with zero padding (i.e., "Ø"). The aligned N-best hypotheses $\mathcal{Y}_N^{ali} = \{Y_1^{ali}, Y_2^{ali}, \cdots, Y_N^{ali}\}$ clearly illustrates the token difference between different candidatures, where each utterance contains $T$ tokens that comes from ASR vocabulary $\mathcal{V}$ plus zero padding Ø:

$$Y_i^{ali} = [y_{i_1}^{ali}, y_{i_2}^{ali}, \cdots, y_{i_T}^{ali}], \quad y_{i_t}^{ali} \in \mathcal{V} \cup \varnothing, \quad (5)$$

Inspired by edit distance, we design an "edit embedding" to capture the token-level difference between two hypotheses, which directly corresponds to their gap in final WER performance. Then, similar to Eq.(4), we calculate the token-level noise embedding by summing up the edit embedding between different pairs of hypotheses in the N-best list:

$$E_{\text{LN}}^{tok} = \text{Concat}\{E_{\text{edit}}(Y_i^{ali}, Y_j^{ali})_{i,j=1,i>j}^N\} \in \mathbb{R}^{\frac{N(N-1)}{2} \times D_{\text{sbert}}},$$

$$E_{\text{edit}}(Y_i^{ali}, Y_j^{ali}) = \sum_{t=1}^{T} [\mathcal{E}_{\text{sbert}}(y_{i_t}^{ali}) - \mathcal{E}_{\text{sbert}}(y_{j_t}^{ali})], \quad (6)$$

Note that we employ SBERT again to extract the token embedding, as it can produce informative embeddings for both utterances and tokens (Reimers & Gurevych, 2019).

## 4.3 AUDIO NOISE DISTILLATION

---

**Algorithm 1** Audio noise distillation via mutual information neural estimation (MINE).

**Require:** LLM $\mathcal{M}_{\text{H2T}}$ with adapter $\mathcal{G}_{\boldsymbol{v}}$, MINE statistics network $\psi$ of parameters $\boldsymbol{\theta}$, language embedding tuner $\mathcal{T}$ of parameters $\boldsymbol{\omega}$. N-best hypotheses $\mathcal{Y}_N$. Parallel noisy speech $\mathcal{X}_n$ and clean speech data $\mathcal{X}_c$. Batch size $B$ and the total number of iterations $M$. Hyper-parameter weight $\lambda$.

1: **for** $m = 1$ **to** $M$ **do**
2:      Draw $B$ N-best hypotheses samples from RobustHP dataset: $\{\mathcal{Y}_N^{(1)}, \mathcal{Y}_N^{(2)}, \cdots, \mathcal{Y}_N^{(B)}\}$;
3:      Draw corresponding noisy and clean speech samples: $\{(X_n^{(1)}, X_c^{(1)}), (X_n^{(2)}, X_c^{(2)}), \cdots, (X_n^{(B)}, X_c^{(B)})\}$;
4:      Extract language-space noise embedding from N-best list using Eq.(4-6): $\{E_{\text{LN}}^{(1)}, E_{\text{LN}}^{(2)}, \cdots, E_{\text{LN}}^{(B)}\}$;
5:      Calculate Eq.(8): $\mathcal{I} = \frac{1}{B} \sum_{b=1}^{B} \psi_{\boldsymbol{\theta}}(E_{\text{LN}}^{(b)}, \mathcal{E}_{\text{ASR}}(X_n^{(b)})) - \log(\frac{1}{B} \sum_{b=1}^{B} e^{\psi_{\boldsymbol{\theta}}(E_{\text{LN}}^{(b)}, \mathcal{E}_{\text{ASR}}(X_c^{(b)}))})$;
6:      Calculate $g_{\boldsymbol{\theta}} = \nabla_{\boldsymbol{\theta}}(\mathcal{I})$ and update $\boldsymbol{\theta}$ by gradient ascent: $\boldsymbol{\theta} \leftarrow \boldsymbol{\theta} + g_{\boldsymbol{\theta}}$;
7:      Calculate GER cost function $\mathcal{L}_{\text{H2T}}$ using Eq.(2), with $\mathcal{T}_{\boldsymbol{\omega}}(E_{\text{LN}}^{(b)})$ incorporated for denoising;
8:      Re-calculate the first term of Eq.(8): $\mathcal{I}_1 = \frac{1}{B} \sum_{b=1}^{B} \psi_{\boldsymbol{\theta}}(\mathcal{T}_{\boldsymbol{\omega}}(E_{\text{LN}}^{(b)}), \mathcal{E}_{\text{ASR}}(X_n^{(b)}))$;
9:      Calculate $g_{\boldsymbol{v}, \boldsymbol{\omega}} = \nabla_{\boldsymbol{v}, \boldsymbol{\omega}}(\mathcal{L}_{\text{H2T}} - \lambda \mathcal{I}_1)$ and update $\boldsymbol{v}, \boldsymbol{\omega}$ by gradient descent: $\boldsymbol{v} \leftarrow \boldsymbol{v} - g_{\boldsymbol{v}}, \boldsymbol{\omega} \leftarrow \boldsymbol{\omega} - g_{\boldsymbol{\omega}}$;
10: **end for**

---

After extracting the language-space noise embedding from N-best hypotheses, we further propose an audio noise distillation approach via mutual information estimation to enhance its noise representation ability. Mutual information (MI) is a measure of dependence between random variables based on the Shannon entropy, which is equivalent to the Kullback-Leibler (KL-) divergence between the joint distribution and the product of the marginal distribution of random variables. Given two random variables $X$ and $Z$, their MI can be calculated by:

$$I(X; Z) = D_{KL}(\mathbb{P}_{XZ} \parallel \mathbb{P}_X \mathbb{P}_Z), \qquad (7)$$

Figure 3: Audio noise distillation by mutual information neural estimation (MINE). The trainable tuner is designed to maximize the MI between our extracted noise embedding and the noisy speech.

where $D_{KL}(\mathbb{P} \parallel \mathbb{Q})$ denotes KL-divergence. However, it is intractable to directly calculate MI based on Eq.(7), so we leverage an estimation method called mutual information neural estimation (MINE) from previous work (Belghazi et al., 2018). MINE employs a statistics network $\psi_{\boldsymbol{\theta}} : \mathcal{X} \times \mathcal{Z} \to \mathbb{R}$ parameterized by $\theta \in \Theta$ to estimate a *neural information measure*:

$$I_{\Theta}(X; Z) = \sup_{\theta \in \Theta} \mathbb{E}_{\mathbb{P}_{XZ}}[\psi_{\boldsymbol{\theta}}] - \log(\mathbb{E}_{\mathbb{P}_X \mathbb{P}_Z}[e^{\psi_{\boldsymbol{\theta}}}]), \qquad (8)$$

In practice, we employ the extracted language-space noise embedding $E_{\text{LN}}$ and noisy audio embedding $\mathcal{E}_{\text{ASR}}(X_n)$ as the joint distribution, while using $E_{\text{LN}}$ and clean audio embedding $\mathcal{E}_{\text{ASR}}(X_c)$ as the marginal distribution, as the noise information only exists in noisy speech.

Algorithm 1 describes how MINE is utilized for audio noise distillation, which includes two stages. First, the statistics network $\psi_{\boldsymbol{\theta}}$ is trained to learn accurate MI estimation using both the positive and negative sample pairs introduced above. Second, a learnable tuner $\mathcal{T}_{\boldsymbol{\omega}}$ is introduced to modulate the language embedding $E_{\text{LN}}$ to capture more real noise information, by maximizing the MI between it and the noisy audio embeddings. More details about the MINE-based audio noise distillation are in §B.2. In addition, the LLM adapter is also updated in second stage to learn H2T mapping for GER.

## 5 EXPERIMENTS

### 5.1 SETUP

We conduct experiments on the proposed RobustHP dataset, which is detailed in §A. To verify the general effectiveness of our approach, we utilize various latest LLMs for evaluation, including LLaMA-2-7b/13b (Touvron et al., 2023b), LLaMA-7b (Touvron et al., 2023a) and Falcon-7b (Penedo et al., 2023). We follow the LLM-Adapter in previous work (Zhang et al., 2023b) for both LLM finetuning and noise embedding incorporation. Details of model and experiment setups are in §C.

Table 1: WER (%) results of RobustGER with LLaMA-2-7b finetuning. "$LM_{rank}$" denotes LM rescoring. "+ Audio Denoising" denotes introducing audio embedding to denoise GER. $o_{nb}$ and $o_{cp}$ respectively denote the N-best oracle and compositional oracle that are defined in §5.1. The subscript percentage denotes relative WER reduction over ASR baseline, i.e., GER improvement.

| Test Set | | Baseline | $LM_{rank}$ | GER | + Audio Denoising | RobustGER (ours) | Oracle $o_{nb}$ | $o_{cp}$ |
|---|---|---|---|---|---|---|---|---|
| CHiME-4 | *test-real* | 12.6 | 12.2 | $6.5_{-48.4\%}$ | $6.4_{-49.2\%}$ | $\mathbf{5.6}_{-55.6\%}$ | 10.5 | 3.0 |
| | *test-simu* | 15.4 | 14.5 | $9.2_{-40.3\%}$ | $9.0_{-41.6\%}$ | $\mathbf{8.2}_{-46.8\%}$ | 12.9 | 5.0 |
| | *dev-real* | 10.6 | 10.3 | $5.0_{-52.8\%}$ | $4.9_{-53.8\%}$ | $\mathbf{4.1}_{-61.3\%}$ | 9.1 | 2.1 |
| | *dev-simu* | 12.4 | 11.9 | $6.8_{-45.2\%}$ | $6.6_{-46.8\%}$ | $\mathbf{5.8}_{-53.2\%}$ | 10.6 | 3.3 |
| | *avg.* | 12.8 | 12.2 | $6.9_{-46.1\%}$ | $6.7_{-47.7\%}$ | $\mathbf{5.9}_{-53.9\%}$ | 10.8 | 3.4 |
| VB-DEMAND | *baby-cry* | 8.0 | 7.8 | $7.0_{-12.5\%}$ | $6.9_{-13.8\%}$ | $\mathbf{6.0}_{-25.0\%}$ | 4.5 | 3.0 |
| | *helicopter* | 8.4 | 8.1 | $7.4_{-11.9\%}$ | $7.3_{-13.1\%}$ | $\mathbf{6.9}_{-17.9\%}$ | 4.8 | 3.2 |
| | *crowd-party* | 22.6 | 22.3 | $21.4_{-5.3\%}$ | $21.0_{-7.1\%}$ | $\mathbf{19.2}_{-15.0\%}$ | 16.5 | 11.5 |
| | *avg.* | 13.0 | 12.7 | $11.9_{-8.5\%}$ | $11.7_{-10.0\%}$ | $\mathbf{10.7}_{-17.7\%}$ | 8.6 | 5.9 |
| NOIZEUS | *babble* | 16.5 | 16.7 | $16.5_{-0.0\%}$ | $16.1_{-2.4\%}$ | $\mathbf{14.5}_{-12.1\%}$ | 9.5 | 5.8 |
| | *car* | 17.4 | 16.8 | $15.3_{-12.1\%}$ | $15.2_{-12.6\%}$ | $\mathbf{14.9}_{-14.4\%}$ | 9.9 | 7.9 |
| | *station* | 12.0 | 11.6 | $10.3_{-14.2\%}$ | $10.3_{-14.2\%}$ | $\mathbf{9.5}_{-20.8\%}$ | 6.6 | 5.0 |
| | *train* | 15.3 | 15.2 | $14.9_{-2.6\%}$ | $15.0_{-2.0\%}$ | $\mathbf{14.9}_{-2.6\%}$ | 10.3 | 7.9 |
| | *street* | 17.4 | 17.2 | $17.4_{-0.0\%}$ | $17.1_{-1.7\%}$ | $\mathbf{16.1}_{-7.5\%}$ | 12.4 | 9.9 |
| | *airport* | 11.2 | 11.0 | $10.7_{-4.5\%}$ | $10.5_{-6.3\%}$ | $\mathbf{9.5}_{-15.2\%}$ | 7.9 | 4.5 |
| | *exhibition* | 13.2 | 13.2 | $12.8_{-3.0\%}$ | $12.4_{-6.1\%}$ | $\mathbf{9.5}_{-28.0\%}$ | 8.3 | 5.8 |
| | *restaurant* | 13.2 | 13.0 | $12.4_{-6.1\%}$ | $12.5_{-5.3\%}$ | $\mathbf{12.0}_{-9.1\%}$ | 8.7 | 6.2 |
| | *avg.* | 14.5 | 14.3 | $13.8_{-4.8\%}$ | $13.6_{-6.2\%}$ | $\mathbf{12.6}_{-13.1\%}$ | 9.2 | 6.6 |
| LS-FreeSound | *metro* | 9.9 | 9.8 | $9.5_{-4.0\%}$ | $9.4_{-5.1\%}$ | $\mathbf{8.9}_{-10.1\%}$ | 7.9 | 4.9 |
| | *car* | 4.0 | 4.0 | $3.7_{-7.5\%}$ | $3.5_{-12.5\%}$ | $\mathbf{3.1}_{-22.5\%}$ | 3.0 | 1.8 |
| | *traffic* | 8.3 | 8.2 | $8.0_{-3.6\%}$ | $7.8_{-6.0\%}$ | $\mathbf{7.5}_{-9.6\%}$ | 6.8 | 4.5 |
| | *cafe* | 9.8 | 9.5 | $8.1_{-17.3\%}$ | $8.1_{-17.3\%}$ | $\mathbf{7.5}_{-23.5\%}$ | 7.1 | 4.6 |
| | *babble* | 32.0 | 31.8 | $31.3_{-2.2\%}$ | $31.6_{-1.3\%}$ | $\mathbf{31.1}_{-2.8\%}$ | 28.7 | 19.3 |
| | *ac/vacuum* | 12.4 | 12.5 | $12.3_{-0.8\%}$ | $12.1_{-2.4\%}$ | $\mathbf{11.4}_{-8.1\%}$ | 10.2 | 6.2 |
| | *avg.* | 12.7 | 12.6 | $12.2_{-3.9\%}$ | $12.1_{-4.7\%}$ | $\mathbf{11.6}_{-8.7\%}$ | 10.6 | 6.9 |
| RATS | *test* | 45.7 | 45.6 | $45.2_{-1.1\%}$ | $44.8_{-2.0\%}$ | $\mathbf{43.2}_{-5.5\%}$ | 38.8 | 23.6 |

Table 2: WER (%) results of RobustGER on different SNR-level testing conditions. The test sets are from LS-FreeSound dataset, with five SNR levels on two noise types. More results are in Table 11.

| Noise Type | SNR (dB) | Baseline | $LM_{rank}$ | GER | + Audio Denoising | RobustGER (ours) | Oracle $o_{nb}$ | $o_{cp}$ |
|---|---|---|---|---|---|---|---|---|
| Metro | 0 | 9.9 | 9.8 | $9.5_{-4.0\%}$ | $9.4_{-5.1\%}$ | $\mathbf{8.9}_{-10.1\%}$ | 7.9 | 4.9 |
| | 5 | 7.2 | 7.0 | $6.7_{-6.9\%}$ | $6.4_{-11.1\%}$ | $\mathbf{5.5}_{-23.6\%}$ | 5.5 | 3.2 |
| | 10 | 4.8 | 4.6 | $4.2_{-12.5\%}$ | $4.3_{-10.4\%}$ | $\mathbf{4.0}_{-16.7\%}$ | 3.9 | 2.3 |
| | 15 | 3.9 | 3.5 | $3.2_{-17.9\%}$ | $3.2_{-17.9\%}$ | $\mathbf{3.0}_{-23.1\%}$ | 3.1 | 1.7 |
| | 20 | 3.3 | 3.1 | $2.7_{-18.2\%}$ | $2.6_{-21.2\%}$ | $\mathbf{2.3}_{-30.3\%}$ | 2.6 | 1.3 |
| | *avg.* | 5.8 | 5.6 | $5.3_{-8.6\%}$ | $5.2_{-10.3\%}$ | $\mathbf{4.7}_{-19.0\%}$ | 4.6 | 2.7 |
| AC/Vacuum | 0 | 12.4 | 12.5 | $12.3_{-0.8\%}$ | $12.1_{-2.4\%}$ | $\mathbf{11.4}_{-8.1\%}$ | 10.2 | 6.2 |
| | 5 | 7.4 | 7.0 | $6.5_{-12.2\%}$ | $6.3_{-14.9\%}$ | $\mathbf{5.8}_{-21.6\%}$ | 5.5 | 3.1 |
| | 10 | 6.6 | 6.2 | $5.5_{-16.7\%}$ | $5.6_{-15.2\%}$ | $\mathbf{5.5}_{-16.7\%}$ | 4.5 | 2.6 |
| | 15 | 4.4 | 4.2 | $3.7_{-15.9\%}$ | $3.7_{-15.9\%}$ | $\mathbf{3.6}_{-18.2\%}$ | 3.3 | 1.8 |
| | 20 | 3.8 | 3.7 | $3.3_{-13.2\%}$ | $3.2_{-15.8\%}$ | $\mathbf{2.9}_{-23.7\%}$ | 2.8 | 1.4 |
| | *avg.* | 6.9 | 6.7 | $6.3_{-8.7\%}$ | $6.2_{-10.1\%}$ | $\mathbf{5.8}_{-15.9\%}$ | 5.3 | 3.0 |
| Clean | $\infty$ | 3.0 | 2.8 | $2.5_{-16.7\%}$ | $2.4_{-20.0\%}$ | $\mathbf{2.1}_{-30.0\%}$ | 2.5 | 1.4 |

We report experimental results in terms of word error rate (WER) and relative GER improvement. We also report two oracle WERs for reference: 1) N-best oracle $o_{nb}$: WER of the "best candidate" in N-best list, and 2) compositional oracle $o_{cp}$: best achievable WER using all the tokens in N-best hypotheses. They indicate the upper-bounds of rerank and GER (using occurred tokens), respectively.

## 5.2 PERFORMANCE OF ROBUSTGER

Table 1 presents the experiment results on LLaMA-2-7b, and more LLMs are evaluated in §D.1. First, we can observe minor gains of performance brought by typical LM rescoring over the Whisper ASR baseline. Compared to LM rescoring, GER achieves promising progress by leveraging LLMs to generate transcription, while its performance gains in most noisy conditions except CHiME-4 are still

Table 3: Ablation study of the language-space noise embedding in terms of utterance and token levels. More studies are presented in Table 13 and Table 14.

| Test Set | | Baseline | GER | + Audio Denoising | + Language Denoising | | |
|---|---|---|---|---|---|---|---|
| | | | | | *Utt.-level* | *Tok.-level* | *Both* |
| CHiME-4 | *test-real* | 12.6 | $6.5_{-48.4\%}$ | $6.4_{-49.2\%}$ | $6.4_{-49.2\%}$ | $6.1_{-51.6\%}$ | $\mathbf{5.9}_{-53.2\%}$ |
| | *test-simu* | 15.4 | $9.2_{-40.3\%}$ | $9.0_{-41.6\%}$ | $9.1_{-40.9\%}$ | $8.9_{-42.2\%}$ | $\mathbf{8.6}_{-44.2\%}$ |
| | *dev-real* | 10.6 | $5.0_{-52.8\%}$ | $4.9_{-53.8\%}$ | $4.7_{-55.7\%}$ | $4.4_{-58.5\%}$ | $\mathbf{4.4}_{-58.5\%}$ |
| | *dev-simu* | 12.4 | $6.8_{-45.2\%}$ | $6.6_{-46.8\%}$ | $6.4_{-48.4\%}$ | $6.3_{-49.2\%}$ | $\mathbf{6.1}_{-50.8\%}$ |
| | *avg.* | 12.8 | $6.9_{-46.1\%}$ | $6.7_{-47.7\%}$ | $6.7_{-47.7\%}$ | $6.4_{-50.0\%}$ | $\mathbf{6.3}_{-50.8\%}$ |
| VB-DEMAND | *baby-cry* | 8.0 | $7.0_{-12.5\%}$ | $6.9_{-13.8\%}$ | $6.7_{-16.3\%}$ | $6.6_{-17.5\%}$ | $\mathbf{6.4}_{-20.0\%}$ |
| | *helicopter* | 8.4 | $7.4_{-11.9\%}$ | $7.3_{-13.1\%}$ | $7.3_{-13.1\%}$ | $7.1_{-15.5\%}$ | $\mathbf{7.1}_{-15.5\%}$ |
| | *crowd-party* | 22.6 | $21.4_{-5.3\%}$ | $21.0_{-7.1\%}$ | $20.8_{-8.0\%}$ | $20.3_{-10.2\%}$ | $\mathbf{19.9}_{-11.9\%}$ |
| | *avg.* | 13.0 | $11.9_{-8.5\%}$ | $11.7_{-10.0\%}$ | $11.6_{-10.8\%}$ | $11.3_{-13.1\%}$ | $\mathbf{11.1}_{-14.6\%}$ |

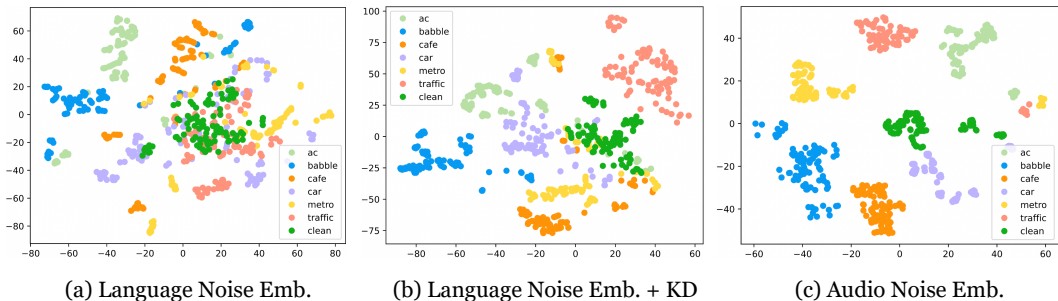

(a) Language Noise Emb.     (b) Language Noise Emb. + KD     (c) Audio Noise Emb.

Figure 4: t-SNE visualizations of (a) language-space noise embedding, (b) language embedding with audio distillation, (c) audio noise embeddings. Cluster distances are in Table 17. Details are in §C.2.

limited. Introducing audio denoising further improves the result but suffers from the cross-modality gap. In comparison, with the proposed language-space denoising approach, our RobustGER achieves significant gains of performance in various noise conditions, with up to 53.9% GER improvement in terms of WER metric, where some results even surpass the reranking upper-bound.

Table 2 reports the performance of RobustGER under different SNRs, where we can observe consistent WER improvements on various noise levels. In addition, RobustGER also shows great effectiveness on clean test data with 30.0% relative WER reduction, which verifies its excellent generality.

## 5.3 ABLATION STUDY

Table 3 illustrates the ablation study on the extraction of language-space noise embedding, which includes both utterance- and token-level information as introduced in §4.2. We can observe that utterance-level embedding only yields minor improvements over vanilla GER, indicating that the global semantics diversity of N-best hypotheses is not fine-grained enough for error correction. On the other hand, token-level information plays a significant role in language-space denoising for GER, as it directly corresponds to the word error rate metric. Combining both performs the best by leveraging richer information to measure N-best list diversity.

In addition, we also conduct ablation studies on the language embedding extractor (i.e., SBERT vs. FastText (Grave et al., 2018), LLaMA embedding.) in §D.3, as well as the audio noise distillation techniques (i.e., MINE vs. contrastive learning, teacher-student learning) in §D.4. All of them verify the effectiveness of our specific designs in RobustGER system.

## 5.4 ANALYSIS

**Visualizations of Noise Embeddings.** Fig. 4 visualizes the language-space noise embedding to show its representativeness of audio noise. First, we can observe from Fig. (a) that our extracted language embedding from the N-best list can well represent some noise types (i.e., "ac", "babble", "cafe"), while the others are intertwined with clean embeddings, indicating less optimal noise representations. For reference, the audio noise embeddings in Fig. (c) distinguish well between different conditions. Therefore, we design a KD approach to distill the real noise information in audio embedding to our

Table 4: Data efficiency of RobustGER on CHiME-4 test sets. The "1k", "2k", etc., denote the number of HT pairs in training data, and "Training Hours" denote its duration of source speech data.

| Test Set | Baseline | GER | RobustGER | | | | |
|---|---|---|---|---|---|---|---|
| | | | 1k | 2k | 5k | 8k | 9.6k (full) |
| Training Hours | - | 17.5 | 1.7 | 3.5 | 9.2 | 14.5 | 17.5 |
| *test-real* | 12.6 | $6.5_{-48.4\%}$ | $9.3_{-26.2\%}$ | $7.0_{-44.4\%}$ | $5.9_{-53.2\%}$ | $5.7_{-54.8\%}$ | $\mathbf{5.6}_{-\mathbf{55.6\%}}$ |
| *test-simu* | 15.4 | $9.2_{-40.3\%}$ | $11.4_{-26.0\%}$ | $9.5_{-38.3\%}$ | $8.8_{-42.9\%}$ | $8.4_{-45.5\%}$ | $\mathbf{8.2}_{-\mathbf{46.8\%}}$ |
| *dev-real* | 10.6 | $5.0_{-52.8\%}$ | $7.2_{-32.1\%}$ | $5.2_{-50.9\%}$ | $4.4_{-58.5\%}$ | $4.1_{-61.3\%}$ | $\mathbf{4.1}_{-\mathbf{61.3\%}}$ |
| *dev-simu* | 12.4 | $6.8_{-45.2\%}$ | $8.9_{-28.2\%}$ | $7.1_{-42.7\%}$ | $6.2_{-50.0\%}$ | $5.9_{-52.4\%}$ | $\mathbf{5.8}_{-\mathbf{53.2\%}}$ |
| *avg.* | 12.8 | $6.9_{-46.1\%}$ | $9.2_{-28.1\%}$ | $7.2_{-43.8\%}$ | $6.3_{-50.8\%}$ | $6.0_{-53.1\%}$ | $\mathbf{5.9}_{-\mathbf{53.9\%}}$ |

Table 5: Case study of RobustGER. We also implement an in-context learning baseline by ChatGPT for comparison (details are in §C.2). The test sample is selected from the CHiME-4 *dev-real* set.

| Method | Utterance | WER (%) |
|---|---|---|
| N-best Candidates | the four other utility company owners will also have to take write ups | 7.7 |
| | the four other utility company owners will also have to take write ups | 7.7 |
| | the four other utility company owners will also have to take write ups | 7.7 |
| | the four other utility company owners will also have to take ride outs | 15.4 |
| | the four other utility company owners will also have to take ride outs | 15.4 |
| In-context Learning | the four other utility company owners will also have to take write-ups | 15.4 |
| GER | the four other utility company owners will also have to take write ups | 7.7 |
| RobustGER | the four other utility company owners will also have to take write offs | **0.0** |
| Ground Truth | the four other utility company owners will also have to take write offs | - |

language embedding. Fig. (b) shows it disentangles the embeddings from different noise conditions and improves their noise representativeness, which leads to better WER results as shown in Table 14.

**Data Efficiency.** As shown in Table 4, we further discuss the data efficiency of RobustGER using the CHiME-4 dataset, whose training set contains 9.6k HT pairs decoded from 17.5-hour speech data. As we gradually reduce the training data, we find that using around half-size data (i.e., 5k pairs) can still maintain the WER performance, i.e., 6.3% vs. 5.9%. When it decreases to 2k pairs, RobustGER is still comparable to GER, i.e., 7.2% vs. 6.9%. This experimental evidence verifies the data efficiency of RobustGER, which may originate from the attribute of parameter-efficient LLM finetuning.

**Case Study.** Table 5 illustrates a case study to demonstrate the effectiveness of RobustGER. There are two errors in N-best hypotheses, i.e., "write ups" (in 1-best) and "ride outs", where the ground truth is "write offs". Both ChatGPT-based in-context learning and LLaMA-based GER fail to correct this error, because the words "write ups" and "write offs" sound quite similar under noisy scenarios. In comparison, our RobustGER can correct this error by *language-space denoising*, where our proposed noise-representative embedding teaches LLMs to remove the language noise in N-best hypotheses that is caused by audio noise. More importantly, the semantic meanings of "write ups" and "write offs" are opposite, which highlights the significance of successful error correction by our RobustGER.

# 6 CONCLUSION

In this paper, we first extend the latest ASR generative error correction benchmark to the most common noisy scenarios in real world, with a proposed RobustHP dataset containing 113K hypotheses-transcription pairs decoded from various noisy ASR corpus. Based on that, we propose RobustGER, a noise-aware generative error correction approach based on LLMs to predict the ground-truth transcription based on N-best hypotheses, where an extracted language-space noise embedding with audio distillation is leveraged to teach LLMs to perform denoising in language space. Extensive experiments on various latest LLMs show that our approach achieves a new breakthrough on RobustHP dataset with up to 53.9% error correction improvement in terms of WER while with limited training data. Further analysis verifies the effectiveness of our proposed language-space embedding to represent audio noise, under which off-the-shelf LLMs show strong ability of *language-space denoising*.

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

# APPENDIX

## A  ROBUST HYPORADISE DATASET DETAILS

Table 6: Robust HyPoradise dataset statistics in terms of number of hypotheses-transcription pairs and average utterance length in various noise domains.

| Source | Domain Category | Training Set | # Pairs | Length | Test Set | # Pairs | Length |
|---|---|---|---|---|---|---|---|
| CHiME-4 | Real-world noise | *tr05-real* | 9,600 | 17.0 | *test-real* | 1,320 | 16.4 |
| | | | | | *test-simu* | 1,320 | 16.4 |
| | | | | | *dev-real* | 1,640 | 16.8 |
| | | | | | *dev-simu* | 1,640 | 16.8 |
| VB-DEMAND | Unseen noise | *train* | 23,075 | 7.5 | *baby-cry* *helicopter* *crowd-party* | 824 | 7.7 |
| NOIZEUS | Real-world noise | *train* | 23,807 | 7.1 | *babble* *car* *station* *train* *street* *airport* *exhibition* *restaurant* | 30 | 8.1 |
| LS-FreeSound | Real-world noise | *train* | 28,539 | 35.0 | *metro* *car* *traffic* *cafe* *babble* *ac/vacuum* | 118 | 17.4 |
| RATS | Radio noise | *train* | 28,504 | 14.2 | *test* | 1,000 | 10.2 |
| Total | | *train* | 113,525 | 16.8 | *test* | 10,340 | 13.7 |

### A.1  ASR SYSTEM

For ASR beam search decoding, we employ Whisper Large-V2 (Radford et al., 2023), one large-scale pre-trained model developed by OpenAI to generate N-best hypotheses, which has been reported with

several competitive and state-of-the-art performance. Whisper model follows the encoder-decoder Transformer (Vaswani et al., 2017) architecture with 1,550 million parameters, which is trained on 680K hours of multilingual and multitask supervised data collected from the web. As a result, it shows universal and excellent noise-robustness in various conditions though lacks of domain specificity (i.e., still lags behind the specifically trained model on certain dataset).

With such pre-trained ASR model, we employ the beam search algorithm for decoding and generate N-best hypotheses list for each speech sample, where the beam size is set to 50. After removing repetitive utterances, we select top-5 hypotheses in terms of posterior probabilities as N-best list. To develop the RobustHP dataset, we carry out this decoding strategy on multiple noisy ASR corpus (see §A.2) and generate data pairs of 5-best hypotheses and ground-truth transcription.

## A.2 SPEECH CORPUS SELECTION

For speech corpus selection, our goal is to cover common noisy ASR scenarios in real world. Consequently, we collect and simulate the following corpus with evident domain characteristics to compose the Robust HyPoradise dataset:

**CHiME-4** (Vincent et al., 2016): CHiME-4 is a popular dataset for far-field noisy speech recognition. It includes real and simulated noisy recordings in four noisy environments, i.e., bus, cafe, pedestrian area, and street junction. We use its *tr05-real* split (9,600 utterances) to generate RobustHP training data, as well as the *test-real* (1,320 utterances), *test-simu* (1,320 utterances), *dev-real* (1,640 utterances) and *dev-simu*(1,640 utterances) splits to generate the test data.

**VoiceBank-DEMAND** (Valentini-Botinhao et al., 2016): VoiceBank-DEMAND is a popular dataset for noisy speech recognition and speech enhancement. We use its training data to build RobustHP, which contains 23,075 noisy utterances from 56 speakers in VoiceBank corpus (Veaux et al., 2013) that are recorded at sampling rate of 16 kHz and mixed with 10 different noise types (babble, cafeteria, car, kitchen, meeting, metro, restaurant, speech-shaped noise, station, traffic) at SNR levels of 0, 5, 10, and 15 dB. For test set, to simulate the challenging unseen noise conditions in practical, we mix the VoiceBank clean test data with three new types of noise (Lin et al., 2021), i.e., baby-cry, helicopter, and crowd-party, at SNR level of 0dB. The test set contains 824 utterances from 2 speakers.

**NOIZEUS** (Hu & Loizou, 2006): NOIZEUS is a noisy speech corpus developed to evaluate noise-robust speech recognition and speech enhancement algorithms. It only contains a test set of 30 IEEE sentences (produced by 3 male and 3 female speakers) corrupted by 8 different real-world noises at SNR levels of 0, 5, 10, and 15 dB, where we select 5 dB for main experiments. The noise was taken from the AURORA-2 database (Hirsch & Pearce, 2000) that includes suburban train noise, babble, car, exhibition hall, restaurant, street, airport and train-station noise. To match the short length of NOIZEUS test utterances (8.1 tokens in average), we select the clean speech from LibriSpeech *train-clean-100* and VoiceBank corpus that with no more than 12 tokens in transcription, and mix them with AURORA-2 noises at SNR levels of 0, 5, 10, 15, and 20 dB to form training set.

**LibriSpeech-FreeSound** (Prasad et al., 2021): LibriSpeech-FreeSound is a simulated noisy speech corpus for noise-robust speech recognition, which mixes the clean speech data from LibriSpeech *train-clean-100* split (Panayotov et al., 2015) and noise data from FreeSound corpus (Font et al., 2013) at SNRs of 0, 5, 10, 15, 20, and 25 dB to form the training set. For test set, they select 118 clean speech samples from LibriSpeech *test-clean* split and mix them with FreeSound noise at SNRs of 0, 5, 10, 15, and 20 dB, where we select 0 dB for main experiments. Six noise types in FreeSound are employed, including metro, car, traffic, cafe, babble and ac/vacuum.

**RATS** (Graff et al., 2014): Robust Automatic Transcription of Speech (RATS) dataset contains radio-communication speech in ultra high frequency data category that is extremely noisy and challenging for ASR task. Its training data contains 43,112 noisy speech utterances, where we filter out the low-quality samples (i.e., WER by Whisper is larger than 0.9) to form the training set. Its test set contains 7,591 utterances, where we randomly select 1,000 samples for higher evaluation efficiency.

## A.3 STATISTICS

After performing beam search decoding on the selected speech corpus introduced above, we collect 113K pairs of N-best hypotheses and ground-truth transcription to form the RobustHP dataset. The statistics are presented in Table 6, which illustrates the number of hypotheses-transcription pairs and

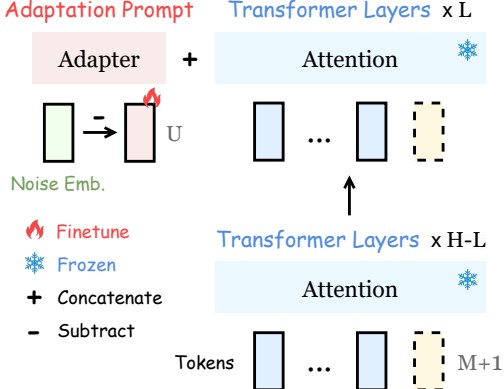

Figure 5: LLaMA-Adapter tuning (Zhang et al., 2023b) with language-space denoising (ours).

the average utterance length in various domains and splits. We would release the RobustHP dataset to public upon publication and open the development venue for more data.

## B   METHOD DETAILS

### B.1   DENOISED LLM FINETUNING

#### B.1.1   EFFICIENT LLM FINETUNING: LLAMA-ADAPTER

As presented in Fig. 5, we employ LLaMA-Adapter (Zhang et al., 2023b) for efficient LLM finetuning. Given pre-trained LLM with a $H$-layer Transformer, it inserts a set of learnable adaptation prompts into the top-$L$ layers that learn high-level semantics. Denote the prompt for $l$-th Transformer layer as $\mathcal{G}_l \in \mathbb{R}^{U \times D}$, where $U$ denotes the prompt length and $D$ denotes the LLM embedding size.

Assume we have $M$ tokens containing instruction and already generated response, i.e., $T_l \in \mathbb{R}^{M \times D}$, where $l$ is the layer index, now we aim to predict the $(M + 1)$-th token as part of response. In order to finetune the entire system, the learnable adaptation prompt is concatenated with $T_l$ as prefix, i.e., $[\mathcal{G}_l; T_l] \in \mathbb{R}^{(U+M) \times D}$. In this case, the instruction knowledge learned by $\mathcal{G}_l$ can guide the $T_l$ to generate the subsequent response under teacher-forcing supervision.

Furthermore, considering the prompt $\mathcal{G}_l$ is randomly initialized and thus may disturb the LLM tuning at early training stages, a zero-initialized attention mechanism is designed to mitigate such disturbance. Suppose the LLM is going to generate the $(M + 1)$-th token based on the prompt $\mathcal{G}_l$ and history tokens $T_l$ at $l$-th layer, and we denote the current $M$-th token as $T_l^{(M)} \in \mathbb{R}^{1 \times D}$. In attention mechanism, there are firstly three projection layers to generate query, key and value, respectively:

$$Q_l = \text{Linear}_q(T_l^{(M)}), \quad K_l = \text{Linear}_k([\mathcal{G}_l; T_l]), \quad V_l = \text{Linear}_v([\mathcal{G}_l; T_l]), \tag{9}$$

Thereafter, the attention score between key and value can be formulated as $A_l = Q_l \cdot K_l / \sqrt{D} \in \mathbb{R}^{1 \times (U+M)}$, which captures the correlation between current token $T_l^{(M)}$ and all $M$ existed tokens $T_l$ as well as the prompt $\mathcal{G}_l$ to predict next token. Therefore, $A_l$ could be split into two parts:

$$A_l = [A_l^{\mathcal{G}}; A_l^T]^T, \tag{10}$$

where $A_l^{\mathcal{G}} \in \mathbb{R}^{U \times 1}$ denotes the attention score of $U$ adaptation prompts and $A_l^T \in \mathbb{R}^{M \times 1}$ denotes that of $M$ history tokens. Since the adaptation prompts are randomly initialized, their attention scores may cast disturbance on next-token prediction in early training stages. To this end, a learnable gating factor $g_l$ with zero initialization is introduced to adaptively control the importance of prompt in attention, by directly multiplied with its softmax weights from Eq.(10):

$$A_l^g = [g_l \cdot \text{softmax}(A_l^{\mathcal{G}}); \ \text{softmax}(A_l^T)]^T, \tag{11}$$

Finally, the attention output of $l$-th Transformer layer can be calculated with a linear projection:

$$O_l^{(M)} = \text{Linear}_o(A_l^g \cdot V_l) \in \mathbb{R}^{1 \times D}, \tag{12}$$

Table 7: Comparison between main configurations of different popular LLMs.

| LLM | LLaMA-2-7b | LLaMA-7b | Falcon-7b | LLaMA-2-13b |
|---|---|---|---|---|
| Number of Transformer Layers $H$ | 32 | 32 | 32 | 40 |
| Number of Attention Heads $N_{\text{head}}$ | 32 | 32 | 71 | 40 |
| Embedding Size $D$ | 4,096 | 4,096 | 4,544 | 5,120 |
| Block Size $B$ | 4,096 | 2,048 | 2,048 | 4,096 |
| Vocabulary Size $V$ | 32,000 | 32,000 | 65,024 | 32,000 |

It is then utilized to predict the next token $T_l^{(M+1)}$ as part of output response. The proposed zero-initialization mechanism achieves an effective trade-off between the pre-trained knowledge of LLM and the learned instructional knowledge through adaptation prompt.

### B.1.2 Denoised Adapter Tuning

Apart from text instructions, LLaMA-Adapter is also capable of generating response based on other modality inputs (Zhang et al., 2023b). However, the cross-modal gap between text and other modalities may affect the finetuning stability and performance (Li et al., 2023b). Therefore, we propose to extract a language-space noise embedding in §4.2 to replace audio embedding for representing the noise conditions of source speech, i.e., $E_{\text{LN}} = [E_{\text{LN}}^{utt}; E_{\text{LN}}^{tok}] \in \mathbb{R}^{N \cdot (N-1) \times D_{\text{sbert}}}$ according to Eq.(9-12), where $N$ denotes N-best list size and $D_{\text{sbert}}$ denotes SBERT embedding size. Then, we incorporate it into LLaMA-Adapter for denoising via element-wise subtraction:

$$\mathcal{G}_l^{\text{dn}} = \mathcal{G}_l - g_l^{\text{dn}} \cdot \mathcal{T}_\omega(E_{\text{LN}}) \in \mathbb{R}^{U \times D}, \quad \text{we set } U = N \cdot (N-1), \tag{13}$$

where $\mathcal{T}_\omega \in \mathbb{R}^{D \times D_{\text{sbert}}}$ denotes the linear projection tuner introduced in §4.3 for audio noise distillation, the subtraction operation denotes "**de**noise". The $g_l^{\text{dn}}$ is a gating factor to control denoising process. Therefore, the resulted $\mathcal{G}_l^{\text{dn}}$ indicates the adaption prompt with language-space denoising, which will replace the $\mathcal{G}_l$ in Eq.(9-12) for adapter tuning.

### B.2 Audio Noise Distillation

As illustrated in §4.3, the key idea of audio noise distillation is to transfer the real noise information in audio embeddings to our extracted language-space noise embedding, in order to enhance its representation ability of audio noise. The approach we propose is based on mutual information neural estimation (MINE) (Belghazi et al., 2018), which can be split into two stages in Algorithm 1. First, we update the MINE to learn MI estimation, by maximizing the MI between language-space noise embedding and noisy audio embeddings and minimizing the MI between language embedding and clean audio embeddings, i.e., audio noise information exists in noisy speech instead of clean speech. Second, we introduce a learnable tuner to modulate the language-space embedding to include more real noise information by maximizing the MI between it and noisy audio embeddings, which is also jointly optimized with LLM finetuning (i.e., the GER cost function $\mathcal{L}_{\text{H2T}}$ as formulated in Eq.(2)).

The rationale we leverage MINE for distillation instead of other techniques like contrastive learning is due to its strong distinguishing ability, which has been verified by recent applications (Zhu et al., 2021; Zhao et al., 2021; Li et al., 2022; Hu et al., 2023e). On the other hand, directly employing techniques like contrastive learning may not work as the language embedding could be far away from the audio-space noisy and clean embeddings, which means the distance between positive and negative samples (i.e., within audio space) is much smaller than the distance between them and the anchor (i.e., between audio and language spaces). Our ablation study in Table 14 also verifies this limitation.

## C  Experimental Setup Details

### C.1  Model Setups

**LLMs.** We select three latest and popular LLMs for evaluation, including LLaMA-2-7b[4] (Touvron et al., 2023b), LLaMA-7b[5] (Touvron et al., 2023a), Falcon-7b[6] (Penedo et al., 2023). In addition, to explore the influence of LLM model size to our approach, we also report some results on LLaMA-2-13b model[7] (Touvron et al., 2023b). Table 7 compares their main configurations.

**Adapter.** We follow the default setting of LLaMA-Adapter (Zhang et al., 2023b)[8,9] with some modifications. The number of tunable Transformer layers $L$ is set to $H-1$, which means all layers except the first one are tunable with inserted prompts. The prompt length $U$ is set to 20 to match the length of $E_{\text{LN}}$ that equals to $N \cdot (N-1)$, where $N$ is the N-best list size set to 5. To extract the language-space noise embedding from N-best hypotheses, we utilize sentence-BERT[10] (Reimers & Gurevych, 2019) whose embedding size $D_{\text{sbert}}$ is 384.

**MINE.** MINE introduces a statistic network $\psi_{\boldsymbol{\theta}}$ that contains a multi-layer perceptron (MLP) and a Sigmoid activation function to estimate a mutual information value between 0 and 1. It receives two inputs including the Whisper-encoded audio embeddings of size 1280 and the language-space noise embedding of size 384, which are first projected to same hidden dimension and added together, and then go through MLP to generate output of size 1. In particular, to incorporate the modulated noise embedding (with same size as LLM embedding, different from the input language embedding of size 384) into MINE, we design an extra interface to receive it as intermediate features on language-space feature branch. The noise embedding tuner contains a linear projection from the SBERT size of 384 to the LLM embedding size as described in §B.1.2.

### C.2  Training and Evaluation Setups

**LLM Finetuning.** The learning rate is set to $10^{-2}$ for CHiME-4 that is relatively small, and set to $5 \times 10^{-3}$ for relatively large datasets including VB-DEMAND, NOIZEUS, LS-FreeSound and RATS. The batch size is set to 4, with accumulation iterations set to 8 (e.g., effective batch size is 32). We train 2 epochs with AdamW optimizer (Loshchilov & Hutter, 2018), with weight decay set to 0.02 and warmup steps set to 20% of one epoch's steps. In addition, MINE is updated using an extra AdamW optimizer with learning rate that is 10% of LLM tuning, where all other configurations keep the same. The hyper-parameter $\lambda$ in Algorithm 1 is set to 0.5. We use 1 NVIDIA A40 GPU for model training, which takes 1.5 hours for CHiME-4, 2.0 hours for VB-DEMAND, 1.6 hours for NOIZEUS, 4.5 hours for LS-FreeSound, and 3.8 hours for RATS, respectively.

**Instruction-following Finetuning.** As presented in Fig. 2, we leverage instruction-following finetuning strategy for GER, where we design an instruction template:

*"Below is the best-hypotheses transcribed from speech recognition system. Please try to revise it using the words which are only included into other-hypothesis, and write the response for the true transcription.### Best-hypothesis:{1-best hypothesis}### Other-hypothesis:{2∼N-best hypotheses}### Response:"*

We find that different instruction templates would have slight impact on the final GER performance, which is an open question for further discussion. In particular, we design some constraints (*e.g.*, only use the words inside N-best hypotheses list for error correction) to control the quality of response and avoid potential LLM hallucinations (Feldman et al., 2023).

**Response Generation.** In the generation stage, we adopt a temperature of 0.2 and top-1 sampling, i.e., greedy search. We observe the over-confidence phenomenon in our experiments (i.e., output

---

[4]`https://huggingface.co/meta-llama/Llama-2-7b-hf`

[5]`https://huggingface.co/yahma/llama-7b-hf`

[6]`https://huggingface.co/tiiuae/falcon-7b`

[7]`https://huggingface.co/meta-llama/Llama-2-13b-hf`

[8]`https://github.com/Lightning-AI/lit-llama/blob/main/lit_llama/adapter.py`

[9]`https://github.com/Lightning-AI/lit-gpt/blob/main/lit_gpt/adapter.py`

[10]`https://huggingface.co/sentence-transformers/all-MiniLM-L6-v2`

Table 8: WER (%) results of RobustGER with LLaMA-7b finetuning. "$LM_{rank}$" denotes LM rescoring. "+ Audio Denoising" denotes introducing audio embedding to denoise GER. $o_{nb}$ and $o_{cp}$ respectively denote the N-best oracle and compositional oracle that are defined in §5.1. The subscript percentage denotes relative WER reduction over ASR baseline, i.e., GER improvement.

| Test Set | | Baseline | $LM_{rank}$ | GER | + Audio Denoising | RobustGER (ours) | Oracle $o_{nb}$ | $o_{cp}$ |
|---|---|---|---|---|---|---|---|---|
| CHiME-4 | *test-real* | 12.6 | 12.2 | $6.8_{-46.0\%}$ | $6.6_{-47.6\%}$ | $\mathbf{5.7}_{-54.8\%}$ | 10.5 | 3.0 |
| | *test-simu* | 15.4 | 14.5 | $10.1_{-34.4\%}$ | $9.7_{-37.0\%}$ | $\mathbf{8.5}_{-44.8\%}$ | 12.9 | 5.0 |
| | *dev-real* | 10.6 | 10.3 | $4.9_{-53.8\%}$ | $4.7_{-55.7\%}$ | $\mathbf{4.0}_{-62.3\%}$ | 9.1 | 2.1 |
| | *dev-simu* | 12.4 | 11.9 | $6.9_{-44.4\%}$ | $6.8_{-45.2\%}$ | $\mathbf{6.3}_{-49.2\%}$ | 10.6 | 3.3 |
| | *avg.* | 12.8 | 12.2 | $7.2_{-43.8\%}$ | $7.0_{-45.3\%}$ | $\mathbf{6.1}_{-52.3\%}$ | 10.8 | 3.4 |
| VB-DEMAND | *baby-cry* | 8.0 | 7.8 | $7.1_{-11.3\%}$ | $7.2_{-10.0\%}$ | $\mathbf{6.5}_{-18.8\%}$ | 4.5 | 3.0 |
| | *helicopter* | 8.4 | 8.1 | $7.3_{-13.1\%}$ | $7.2_{-14.3\%}$ | $\mathbf{6.8}_{-19.0\%}$ | 4.8 | 3.2 |
| | *crowd-party* | 22.6 | 22.3 | $21.5_{-4.9\%}$ | $21.1_{-6.6\%}$ | $\mathbf{20.1}_{-11.1\%}$ | 16.5 | 11.5 |
| | *avg.* | 13.0 | 12.7 | $12.0_{-7.7\%}$ | $11.8_{-9.2\%}$ | $\mathbf{11.1}_{-14.6\%}$ | 8.6 | 5.9 |
| NOIZEUS | *babble* | 16.5 | 16.7 | $15.3_{-7.3\%}$ | $15.0_{-9.1\%}$ | $\mathbf{13.6}_{-17.6\%}$ | 9.5 | 5.8 |
| | *car* | 17.4 | 16.8 | $14.9_{-14.4\%}$ | $14.8_{-14.9\%}$ | $\mathbf{14.9}_{-14.4\%}$ | 9.9 | 7.9 |
| | *station* | 12.0 | 11.6 | $10.7_{-10.8\%}$ | $10.7_{-10.8\%}$ | $\mathbf{10.3}_{-14.2\%}$ | 6.6 | 5.0 |
| | *train* | 15.3 | 15.2 | $14.5_{-5.2\%}$ | $14.2_{-7.2\%}$ | $\mathbf{12.8}_{-16.3\%}$ | 10.3 | 7.9 |
| | *street* | 17.4 | 17.2 | $16.9_{-2.9\%}$ | $16.7_{-4.0\%}$ | $\mathbf{16.1}_{-7.5\%}$ | 12.4 | 9.9 |
| | *airport* | 11.2 | 11.0 | $10.3_{-8.0\%}$ | $10.1_{-9.8\%}$ | $\mathbf{9.5}_{-15.2\%}$ | 7.9 | 4.5 |
| | *exhibition* | 13.2 | 13.2 | $13.2_{-0.0\%}$ | $13.0_{-1.5\%}$ | $\mathbf{12.8}_{-3.0\%}$ | 8.3 | 5.8 |
| | *restaurant* | 13.2 | 13.0 | $13.6_{+3.0\%}$ | $13.2_{-0.0\%}$ | $\mathbf{12.0}_{-9.1\%}$ | 8.7 | 6.2 |
| | *avg.* | 14.5 | 14.3 | $13.7_{-5.5\%}$ | $13.5_{-6.9\%}$ | $\mathbf{12.8}_{-11.7\%}$ | 9.2 | 6.6 |
| LS-FreeSound | *metro* | 9.9 | 9.8 | $9.4_{-5.1\%}$ | $9.2_{-7.1\%}$ | $\mathbf{8.2}_{-17.2\%}$ | 7.9 | 4.9 |
| | *car* | 4.0 | 4.0 | $3.5_{-12.5\%}$ | $3.6_{-10.0\%}$ | $\mathbf{3.3}_{-17.5\%}$ | 3.0 | 1.8 |
| | *traffic* | 8.3 | 8.2 | $8.3_{-0.0\%}$ | $8.3_{-0.0\%}$ | $\mathbf{8.2}_{-1.2\%}$ | 6.8 | 4.5 |
| | *cafe* | 9.8 | 9.5 | $9.3_{-5.1\%}$ | $9.1_{-7.1\%}$ | $\mathbf{8.5}_{-13.3\%}$ | 7.1 | 4.6 |
| | *babble* | 32.0 | 31.8 | $31.7_{-0.9\%}$ | $31.4_{-1.9\%}$ | $\mathbf{30.9}_{-3.4\%}$ | 28.7 | 19.3 |
| | *ac/vacuum* | 12.4 | 12.5 | $11.8_{-4.8\%}$ | $11.6_{-6.5\%}$ | $\mathbf{11.2}_{-9.7\%}$ | 10.2 | 6.2 |
| | *avg.* | 12.7 | 12.6 | $12.3_{-3.1\%}$ | $12.2_{-3.9\%}$ | $\mathbf{11.7}_{-7.9\%}$ | 10.6 | 6.9 |
| RATS | *test* | 45.7 | 45.6 | $45.5_{-0.4\%}$ | $45.2_{-1.1\%}$ | $\mathbf{43.6}_{-4.6\%}$ | 38.8 | 23.6 |

probability distribution for decision is close to one-hot), which results in similar performance with different $k$ for top-$k$ sampling. Therefore, we select top-1 sampling for higher decoding efficiency.

**LM Rescoring Baseline.** For $LM_{rank}$ baseline in Table 1, we use a Transformer-based LM for typical rescoring, which is trained on the text transcriptions of each RobustHP subset using ESPnet toolkit [11] (Watanabe et al., 2018). The LM contains 16 Transformer layers with 8 heads and 512 attention units, and it is trained for 25 epochs with Adam optimizer (Kingma & Ba, 2014). The learning rate is set to $5 \times 10^{-3}$ with 25,000 warm-up steps.

**In-context Learning Baseline.** We implement an in-context learning baseline for case study in Table 5, which is effective in making full use of LLM's powerful reasoning ability and linguistic knowledge (Dong et al., 2022). In particular, we utilize ChatGPT to conduct GER task using task-activated prompting (TAP) (Yang et al., 2023a): we first prompt ChatGPT to summarize what is ASR and typical LM rescoring, and then inform it the definition of ASR generative error correction, followed by several examples to teach it how to do such kind of error correction. With above background knowledge, we finally ask it to perform GER for our sample in case study.

**Details of t-SNE Visualization.** Fig. 4 and 6 present the t-SNE visualization of the language and audio noise embeddings. The language embeddings are the outputs of distillation tuner, which are selected from the LS-FreeSound test samples. The audio embeddings are encoder outputs of Whisper ASR model, where the speech samples also come from LS-FreeSound test samples. In particular, for better visualization we employ Stable-Whisper[12] to extract the speech segments of same word "for" (i.e., around 5.7s in total from LS-FreeSound test data), as the distance between different phonemes is much larger than that between different noise conditions.

---

[11] https://github.com/espnet/espnet/tree/master/egs2/librispeech/asr1
[12] https://github.com/jianfch/stable-ts

Table 9: WER (%) results of RobustGER with Falcon-7b finetuning. "$LM_{rank}$" denotes LM rescoring. "+ Audio Denoising" denotes introducing audio embedding to denoise GER. $o_{nb}$ and $o_{cp}$ respectively denote the N-best oracle and compositional oracle that are defined in §5.1.

| Test Set | | Baseline | $LM_{rank}$ | GER | + Audio Denoising | RobustGER (ours) | Oracle $o_{nb}$ | $o_{cp}$ |
|---|---|---|---|---|---|---|---|---|
| CHiME-4 | test-real | 12.6 | 12.2 | $7.4_{-41.3\%}$ | $7.2_{-42.9\%}$ | $\mathbf{6.2}_{-50.8\%}$ | 10.5 | 3.0 |
| | test-simu | 15.4 | 14.5 | $10.2_{-33.8\%}$ | $10.0_{-35.1\%}$ | $\mathbf{8.9}_{-42.2\%}$ | 12.9 | 5.0 |
| | dev-real | 10.6 | 10.3 | $5.8_{-45.3\%}$ | $5.5_{-48.1\%}$ | $\mathbf{4.8}_{-54.7\%}$ | 9.1 | 2.1 |
| | dev-simu | 12.4 | 11.9 | $7.7_{-37.9\%}$ | $7.4_{-41.7\%}$ | $\mathbf{6.5}_{-47.6\%}$ | 10.6 | 3.3 |
| | avg. | 12.8 | 12.2 | $7.8_{-39.1\%}$ | $7.5_{-41.4\%}$ | $\mathbf{6.6}_{-48.4\%}$ | 10.8 | 3.4 |
| VB-DEMAND | baby-cry | 8.0 | 7.8 | $7.2_{-10.0\%}$ | $7.0_{-12.5\%}$ | $\mathbf{6.7}_{-16.3\%}$ | 4.5 | 3.0 |
| | helicopter | 8.4 | 8.1 | $7.8_{-7.1\%}$ | $7.7_{-8.3\%}$ | $\mathbf{7.2}_{-14.3\%}$ | 4.8 | 3.2 |
| | crowd-party | 22.6 | 22.3 | $21.7_{-4.0\%}$ | $21.4_{-5.3\%}$ | $\mathbf{20.5}_{-9.3\%}$ | 16.5 | 11.5 |
| | avg. | 13.0 | 12.7 | $12.2_{-6.2\%}$ | $12.0_{-7.7\%}$ | $\mathbf{11.5}_{-11.5\%}$ | 8.6 | 5.9 |
| NOIZEUS | babble | 16.5 | 16.7 | $16.9_{+2.4\%}$ | $16.5_{-0.0\%}$ | $\mathbf{15.3}_{-7.3\%}$ | 9.5 | 5.8 |
| | car | 17.4 | 16.8 | $15.7_{-9.8\%}$ | $15.4_{-11.5\%}$ | $\mathbf{14.9}_{-14.4\%}$ | 9.9 | 7.9 |
| | station | 12.0 | 11.6 | $11.6_{-3.3\%}$ | $11.2_{-6.7\%}$ | $\mathbf{9.1}_{-24.2\%}$ | 6.6 | 5.0 |
| | train | 15.3 | 15.2 | $16.5_{+7.8\%}$ | $14.6_{-4.6\%}$ | $\mathbf{12.8}_{-16.3\%}$ | 10.3 | 7.9 |
| | street | 17.4 | 17.2 | $16.1_{-7.5\%}$ | $\mathbf{16.0}_{-8.0\%}$ | $16.1_{-7.5\%}$ | 12.4 | 9.9 |
| | airport | 11.2 | 11.0 | $10.7_{-4.5\%}$ | $10.6_{-5.4\%}$ | $\mathbf{10.3}_{-8.0\%}$ | 7.9 | 4.5 |
| | exhibition | 13.2 | 13.2 | $12.8_{-3.0\%}$ | $12.5_{-5.3\%}$ | $\mathbf{12.0}_{-9.1\%}$ | 8.3 | 5.8 |
| | restaurant | 13.2 | 13.0 | $12.8_{-3.0\%}$ | $12.6_{-4.5\%}$ | $\mathbf{12.0}_{-9.1\%}$ | 8.7 | 6.2 |
| | avg. | 14.5 | 14.3 | $14.1_{-2.8\%}$ | $13.7_{-5.5\%}$ | $\mathbf{12.8}_{-11.7\%}$ | 9.2 | 6.6 |
| LS-FreeSound | metro | 9.9 | 9.8 | $10.3_{+4.0\%}$ | $9.9_{-0.0\%}$ | $\mathbf{8.9}_{-10.1\%}$ | 7.9 | 4.9 |
| | car | 4.0 | 4.0 | $3.7_{-7.5\%}$ | $3.7_{-7.5\%}$ | $\mathbf{3.5}_{-12.5\%}$ | 3.0 | 1.8 |
| | traffic | 8.3 | 8.2 | $8.2_{-1.2\%}$ | $8.0_{-3.6\%}$ | $\mathbf{7.5}_{-9.6\%}$ | 6.8 | 4.5 |
| | cafe | 9.8 | 9.5 | $8.1_{-17.3\%}$ | $8.0_{-18.4\%}$ | $\mathbf{7.9}_{-19.4\%}$ | 7.1 | 4.6 |
| | babble | 32.0 | 31.8 | $31.1_{-2.8\%}$ | $30.9_{-3.4\%}$ | $\mathbf{30.5}_{-4.7\%}$ | 28.7 | 19.3 |
| | ac/vacuum | 12.4 | 12.5 | $12.6_{+1.6\%}$ | $12.6_{+1.6\%}$ | $\mathbf{12.2}_{-1.6\%}$ | 10.2 | 6.2 |
| | avg. | 12.7 | 12.6 | $12.3_{-3.1\%}$ | $12.2_{-3.9\%}$ | $\mathbf{11.8}_{-7.1\%}$ | 10.6 | 6.9 |
| RATS | test | 45.7 | 45.6 | $45.3_{-0.9\%}$ | $44.9_{-1.8\%}$ | $\mathbf{43.3}_{-5.3\%}$ | 38.8 | 23.6 |

Table 10: WER (%) results of RobustGER with LLaMA-2-13b finetuning. "$LM_{rank}$" denotes LM rescoring. "+ Audio Denoising" denotes introducing audio embedding to denoise GER. $o_{nb}$ and $o_{cp}$ respectively denote the N-best oracle and compositional oracle that are defined in §5.1.

| Test Set | | Baseline | $LM_{rank}$ | GER | + Audio Denoising | RobustGER (ours) | Oracle $o_{nb}$ | $o_{cp}$ |
|---|---|---|---|---|---|---|---|---|
| CHiME-4 | test-real | 12.6 | 12.2 | $5.5_{-56.3\%}$ | $5.3_{-57.9\%}$ | $\mathbf{4.9}_{-61.1\%}$ | 10.5 | 3.0 |
| | test-simu | 15.4 | 14.5 | $8.1_{-47.4\%}$ | $8.2_{-46.8\%}$ | $\mathbf{7.9}_{-48.7\%}$ | 12.9 | 5.0 |
| | dev-real | 10.6 | 10.3 | $4.1_{-61.3\%}$ | $3.8_{-64.2\%}$ | $\mathbf{3.3}_{-68.9\%}$ | 9.1 | 2.1 |
| | dev-simu | 12.4 | 11.9 | $6.1_{-50.8\%}$ | $5.9_{-52.4\%}$ | $\mathbf{5.1}_{-58.9\%}$ | 10.6 | 3.3 |
| | avg. | 12.8 | 12.2 | $6.0_{-53.1\%}$ | $5.8_{-54.7\%}$ | $\mathbf{5.3}_{-58.6\%}$ | 10.8 | 3.4 |
| VB-DEMAND | baby-cry | 8.0 | 7.8 | $6.7_{-16.3\%}$ | $6.6_{-17.5\%}$ | $\mathbf{6.0}_{-25.0\%}$ | 4.5 | 3.0 |
| | helicopter | 8.4 | 8.1 | $7.2_{-14.3\%}$ | $7.0_{-16.7\%}$ | $\mathbf{6.5}_{-22.6\%}$ | 4.8 | 3.2 |
| | crowd-party | 22.6 | 22.3 | $21.0_{-7.1\%}$ | $20.6_{-8.8\%}$ | $\mathbf{19.6}_{-13.3\%}$ | 16.5 | 11.5 |
| | avg. | 13.0 | 12.7 | $11.6_{-10.8\%}$ | $11.4_{-12.3\%}$ | $\mathbf{10.7}_{-17.7\%}$ | 8.6 | 5.9 |
| NOIZEUS | babble | 16.5 | 16.7 | $15.3_{-7.3\%}$ | $\mathbf{15.2}_{-7.9\%}$ | $15.3_{-7.3\%}$ | 9.5 | 5.8 |
| | car | 17.4 | 16.8 | $14.9_{-14.4\%}$ | $14.7_{-15.5\%}$ | $\mathbf{14.0}_{-19.5\%}$ | 9.9 | 7.9 |
| | station | 12.0 | 11.6 | $9.5_{-20.8\%}$ | $9.4_{-21.7\%}$ | $\mathbf{9.1}_{-24.2\%}$ | 6.6 | 5.0 |
| | train | 15.3 | 15.2 | $15.3_{-0.0\%}$ | $14.7_{-3.9\%}$ | $\mathbf{12.8}_{-16.3\%}$ | 10.3 | 7.9 |
| | street | 17.4 | 17.2 | $\mathbf{16.9}_{-2.9\%}$ | $\mathbf{16.9}_{-2.9\%}$ | $16.9_{-2.9\%}$ | 12.4 | 9.9 |
| | airport | 11.2 | 11.0 | $10.7_{-4.5\%}$ | $10.3_{-8.0\%}$ | $\mathbf{8.7}_{-22.3\%}$ | 7.9 | 4.5 |
| | exhibition | 13.2 | 13.2 | $12.0_{-9.1\%}$ | $11.6_{-12.1\%}$ | $\mathbf{10.7}_{-18.9\%}$ | 8.3 | 5.8 |
| | restaurant | 13.2 | 13.0 | $12.4_{-6.1\%}$ | $12.1_{-8.3\%}$ | $\mathbf{10.3}_{-22.0\%}$ | 8.7 | 6.2 |
| | avg. | 14.5 | 14.3 | $13.4_{-7.6\%}$ | $13.1_{-9.7\%}$ | $\mathbf{12.2}_{-15.9\%}$ | 9.2 | 6.6 |
| LS-FreeSound | metro | 9.9 | 9.8 | $9.7_{-2.0\%}$ | $9.4_{-5.1\%}$ | $\mathbf{8.6}_{-13.1\%}$ | 7.9 | 4.9 |
| | car | 4.0 | 4.0 | $3.7_{-7.5\%}$ | $3.8_{-5.0\%}$ | $\mathbf{3.5}_{-12.5\%}$ | 3.0 | 1.8 |
| | traffic | 8.3 | 8.2 | $8.3_{-0.0\%}$ | $8.2_{-1.2\%}$ | $\mathbf{7.6}_{-8.4\%}$ | 6.8 | 4.5 |
| | cafe | 9.8 | 9.5 | $8.7_{-11.2\%}$ | $8.5_{-13.3\%}$ | $\mathbf{7.5}_{-23.5\%}$ | 7.1 | 4.6 |
| | babble | 32.0 | 31.8 | $31.8_{-0.6\%}$ | $31.6_{-1.3\%}$ | $\mathbf{30.8}_{-3.8\%}$ | 28.7 | 19.3 |
| | ac/vacuum | 12.4 | 12.5 | $11.5_{-7.3\%}$ | $11.4_{-8.1\%}$ | $\mathbf{11.0}_{-11.3\%}$ | 10.2 | 6.2 |
| | avg. | 12.7 | 12.6 | $12.3_{-3.1\%}$ | $12.2_{-3.9\%}$ | $\mathbf{11.5}_{-9.4\%}$ | 10.6 | 6.9 |
| RATS | test | 45.7 | 45.6 | $44.4_{-2.8\%}$ | $44.0_{-3.7\%}$ | $\mathbf{43.0}_{-5.9\%}$ | 38.8 | 23.6 |

Table 11: WER (%) results of RobustGER on different SNR-level testing conditions. The test sets are from LS-FreeSound dataset, with five SNR levels (i.e., {0, 5, 10, 15, 20}dB) on six noise types (i.e., "Metro", "Car", "Traffic", "Cafe", "Babble", and "AC/Vacuum").

| Noise Type | SNR (dB) | Baseline | $LM_{rank}$ | GER | + Audio Denoising | RobustGER (ours) | Oracle $o_{nb}$ | $o_{cp}$ |
|---|---|---|---|---|---|---|---|---|
| Metro | 0 | 9.9 | 9.8 | $9.5_{-4.0\%}$ | $9.4_{-5.1\%}$ | $\mathbf{8.9}_{-10.1\%}$ | 7.9 | 4.9 |
| | 5 | 7.2 | 7.0 | $6.7_{-6.9\%}$ | $6.4_{-11.1\%}$ | $\mathbf{5.5}_{-23.6\%}$ | 5.5 | 3.2 |
| | 10 | 4.8 | 4.6 | $4.2_{-12.5\%}$ | $4.3_{-10.4\%}$ | $\mathbf{4.0}_{-16.7\%}$ | 3.9 | 2.3 |
| | 15 | 3.9 | 3.5 | $3.2_{-17.9\%}$ | $3.2_{-17.9\%}$ | $\mathbf{3.0}_{-23.1\%}$ | 3.1 | 1.7 |
| | 20 | 3.3 | 3.1 | $2.7_{-18.2\%}$ | $2.6_{-21.2\%}$ | $\mathbf{2.3}_{-30.3\%}$ | 2.6 | 1.3 |
| | avg. | 5.8 | 5.6 | $5.3_{-8.6\%}$ | $5.2_{-10.3\%}$ | $\mathbf{4.7}_{-19.0\%}$ | 4.6 | 2.7 |
| Car | 0 | 4.0 | 4.0 | $3.7_{-7.5\%}$ | $3.5_{-12.5\%}$ | $\mathbf{3.1}_{-22.5\%}$ | 3.0 | 1.8 |
| | 5 | 3.8 | 3.5 | $3.1_{-18.4\%}$ | $3.1_{-18.4\%}$ | $\mathbf{2.8}_{-26.3\%}$ | 2.8 | 1.5 |
| | 10 | 3.2 | 3.3 | $3.2_{-0.0\%}$ | $3.0_{-6.3\%}$ | $\mathbf{2.2}_{-31.3\%}$ | 2.4 | 1.4 |
| | 15 | 2.8 | 2.7 | $2.5_{-10.7\%}$ | $2.5_{-10.7\%}$ | $\mathbf{2.3}_{-17.9\%}$ | 2.4 | 1.4 |
| | 20 | 3.1 | 2.8 | $2.5_{-19.4\%}$ | $2.4_{-22.6\%}$ | $\mathbf{2.1}_{-32.3\%}$ | 2.4 | 1.4 |
| | avg. | 3.4 | 3.3 | $3.0_{-11.8\%}$ | $2.9_{-14.7\%}$ | $\mathbf{2.5}_{-26.5\%}$ | 2.6 | 1.5 |
| Traffic | 0 | 8.3 | 8.2 | $8.0_{-3.6\%}$ | $7.8_{-6.0\%}$ | $\mathbf{7.5}_{-9.6\%}$ | 6.8 | 4.5 |
| | 5 | 6.3 | 6.1 | $5.6_{-11.1\%}$ | $5.5_{-12.7\%}$ | $\mathbf{4.9}_{-22.2\%}$ | 4.9 | 3.2 |
| | 10 | 3.8 | 3.6 | $3.3_{-13.2\%}$ | $3.3_{-13.2\%}$ | $\mathbf{3.2}_{-15.8\%}$ | 3.2 | 1.9 |
| | 15 | 3.4 | 3.1 | $2.9_{-14.7\%}$ | $2.8_{-17.6\%}$ | $\mathbf{2.4}_{-29.4\%}$ | 2.8 | 1.7 |
| | 20 | 3.7 | 3.5 | $3.4_{-8.1\%}$ | $3.3_{-10.8\%}$ | $\mathbf{3.0}_{-18.9\%}$ | 2.9 | 1.7 |
| | avg. | 5.1 | 4.9 | $4.6_{-9.8\%}$ | $4.5_{-11.8\%}$ | $\mathbf{4.2}_{-17.6\%}$ | 4.1 | 2.6 |
| Cafe | 0 | 9.8 | 9.5 | $8.1_{-17.3\%}$ | $8.1_{-17.3\%}$ | $\mathbf{7.5}_{-23.5\%}$ | 7.1 | 4.6 |
| | 5 | 5.7 | 5.7 | $5.4_{-5.3\%}$ | $5.6_{-1.8\%}$ | $\mathbf{5.3}_{-7.0\%}$ | 4.5 | 2.6 |
| | 10 | 5.0 | 4.7 | $4.5_{-10.0\%}$ | $4.2_{-16.0\%}$ | $\mathbf{4.0}_{-20.0\%}$ | 3.8 | 2.2 |
| | 15 | 3.6 | 3.5 | $3.3_{-8.3\%}$ | $3.2_{-11.1\%}$ | $\mathbf{3.0}_{-16.7\%}$ | 2.7 | 1.5 |
| | 20 | 3.5 | 3.2 | $2.7_{-22.9\%}$ | $2.9_{-17.1\%}$ | $\mathbf{2.9}_{-17.1\%}$ | 2.6 | 1.5 |
| | avg. | 5.5 | 5.3 | $4.8_{-12.7\%}$ | $4.8_{-12.7\%}$ | $\mathbf{4.5}_{-18.2\%}$ | 4.1 | 2.5 |
| Babble | 0 | 32.0 | 31.8 | $31.3_{-2.2\%}$ | $31.6_{-1.3\%}$ | $\mathbf{31.1}_{-2.8\%}$ | 28.7 | 19.3 |
| | 5 | 17.0 | 16.8 | $17.0_{-0.0\%}$ | $16.6_{-2.4\%}$ | $\mathbf{16.3}_{-4.1\%}$ | 13.9 | 9.2 |
| | 10 | 8.8 | 9.0 | $8.6_{-2.3\%}$ | $8.4_{-4.5\%}$ | $\mathbf{8.1}_{-8.0\%}$ | 6.5 | 3.9 |
| | 15 | 6.5 | 6.1 | $5.8_{-10.8\%}$ | $5.7_{-12.3\%}$ | $\mathbf{5.4}_{-16.9\%}$ | 4.7 | 3.0 |
| | 20 | 10.5 | 10.1 | $7.6_{-27.6\%}$ | $7.6_{-27.6\%}$ | $\mathbf{7.6}_{-27.6\%}$ | 9.6 | 2.0 |
| | avg. | 15.0 | 14.8 | $14.1_{-6.0\%}$ | $14.0_{-6.7\%}$ | $\mathbf{13.7}_{-8.7\%}$ | 12.7 | 7.5 |
| AC/Vacuum | 0 | 12.4 | 12.5 | $12.3_{-0.8\%}$ | $12.1_{-2.4\%}$ | $\mathbf{11.4}_{-8.1\%}$ | 10.2 | 6.2 |
| | 5 | 7.4 | 7.0 | $6.5_{-12.2\%}$ | $6.3_{-14.9\%}$ | $\mathbf{5.8}_{-21.6\%}$ | 5.5 | 3.1 |
| | 10 | 6.6 | 6.2 | $5.5_{-16.7\%}$ | $5.6_{-15.2\%}$ | $\mathbf{5.5}_{-16.7\%}$ | 4.5 | 2.6 |
| | 15 | 4.4 | 4.2 | $3.7_{-15.9\%}$ | $3.7_{-15.9\%}$ | $\mathbf{3.6}_{-18.2\%}$ | 3.3 | 1.8 |
| | 20 | 3.8 | 3.7 | $3.3_{-13.2\%}$ | $3.2_{-15.8\%}$ | $\mathbf{2.9}_{-23.7\%}$ | 2.8 | 1.4 |
| | avg. | 6.9 | 6.7 | $6.3_{-8.7\%}$ | $6.2_{-10.1\%}$ | $\mathbf{5.8}_{-15.9\%}$ | 5.3 | 3.0 |
| Clean | ∞ | 3.0 | 2.8 | $2.5_{-16.7\%}$ | $2.4_{-20.0\%}$ | $\mathbf{2.1}_{-30.0\%}$ | 2.5 | 1.4 |

Table 12: WER (%) results of RobustGER on clean test data from VB-DEMAND and LS-FreeSound.

| Test set | Baseline | $LM_{rank}$ | GER | + Audio Denoising | RobustGER (ours) | Oracle $o_{nb}$ | $o_{cp}$ |
|---|---|---|---|---|---|---|---|
| VB-DEMAND | 1.3 | 1.5 | $1.3_{-0.0\%}$ | $1.2_{-7.7\%}$ | $\mathbf{0.7}_{-46.2\%}$ | 0.6 | 0.3 |
| LS-FreeSound | 3.0 | 2.8 | $2.5_{-16.7\%}$ | $2.4_{-20.0\%}$ | $\mathbf{2.1}_{-30.0\%}$ | 2.5 | 1.4 |

## D  SUPPLEMENTARY EXPERIMENTS

### D.1  RESULTS ON DIFFERENT LLMS

Apart from LLaMA-2-7b, we also evaluate our proposed RobustGER approach on popular LLaMA-7b and Falcon-7b models as illustrated in Table 8 and 9. To further investigate the effect of LLM size on RobustGER, we conduct extra experiments on LLaMA-2-13b in Table 10.

Similar to the results of LLaMA-2-7b in Table 1, our proposed RobustGER achieves consistent gains of performance on various LLMs and testing conditions, which verifies its general effectiveness. On the other hand, there exists some performance difference between different LLMs. In particular, LLaMA-2-13b outperforms all the 7b LLMs due to its larger model capacity and stronger language

Table 13: Ablation study of the language-space noise embedding in terms of text embedding extractor. "*LLaMA Emb.*" denotes the input embedding layer of LLaMA-2-7b model.

| Test Set | | Baseline | GER | + Audio Denoising | + Language Denoising | | |
| --- | --- | --- | --- | --- | --- | --- | --- |
| | | | | | *LLaMA Emb.* | *FastText* | *SBERT* |
| CHiME-4 | *test-real* | 12.6 | $6.5_{-48.4\%}$ | $6.4_{-49.2\%}$ | $6.6_{-47.6\%}$ | $6.2_{-50.8\%}$ | $\mathbf{5.9}_{-53.2\%}$ |
| | *test-simu* | 15.4 | $9.2_{-40.3\%}$ | $9.0_{-41.6\%}$ | $8.9_{-42.2\%}$ | $8.7_{-43.5\%}$ | $\mathbf{8.6}_{-44.2\%}$ |
| | *dev-real* | 10.6 | $5.0_{-52.8\%}$ | $4.9_{-53.8\%}$ | $4.9_{-53.8\%}$ | $4.5_{-57.5\%}$ | $\mathbf{4.4}_{-58.5\%}$ |
| | *dev-simu* | 12.4 | $6.8_{-45.2\%}$ | $6.6_{-46.8\%}$ | $6.7_{-46.0\%}$ | $6.4_{-48.4\%}$ | $\mathbf{6.1}_{-50.8\%}$ |
| | *avg.* | 12.8 | $6.9_{-46.1\%}$ | $6.7_{-47.7\%}$ | $6.8_{-46.9\%}$ | $6.5_{-49.2\%}$ | $\mathbf{6.3}_{-50.8\%}$ |
| VB-DEMAND | *baby-cry* | 8.0 | $7.0_{-12.5\%}$ | $6.9_{-13.8\%}$ | $6.8_{-15.0\%}$ | $6.5_{-18.8\%}$ | $\mathbf{6.4}_{-20.0\%}$ |
| | *helicopter* | 8.4 | $7.4_{-11.9\%}$ | $7.3_{-13.1\%}$ | $7.5_{-10.7\%}$ | $7.4_{-11.9\%}$ | $\mathbf{7.1}_{-15.5\%}$ |
| | *crowd-party* | 22.6 | $21.4_{-5.3\%}$ | $21.0_{-7.1\%}$ | $20.9_{-7.5\%}$ | $20.3_{-10.2\%}$ | $\mathbf{19.9}_{-11.9\%}$ |
| | *avg.* | 13.0 | $11.9_{-8.5\%}$ | $11.7_{-10.0\%}$ | $11.7_{-10.0\%}$ | $11.4_{-12.3\%}$ | $\mathbf{11.1}_{-14.6\%}$ |

Table 14: Comparison of different techniques for audio noise distillation. "*T-S Learning*" denotes teacher-student learning with KL regularization, "*Contra. Learning*" denotes contrastive learning.

| Test Set | | Baseline | GER | + Lang. Denoising | + Audio Noise Distillation | | |
| --- | --- | --- | --- | --- | --- | --- | --- |
| | | | | | *T-S learning* | *Contra. learning* | *MINE* |
| CHiME-4 | *test-real* | 12.6 | $6.5_{-48.4\%}$ | $5.9_{-53.2\%}$ | $5.9_{-53.2\%}$ | $5.8_{-54.0\%}$ | $\mathbf{5.6}_{-55.6\%}$ |
| | *test-simu* | 15.4 | $9.2_{-40.3\%}$ | $8.6_{-44.2\%}$ | $8.7_{-43.5\%}$ | $8.4_{-45.5\%}$ | $\mathbf{8.2}_{-46.8\%}$ |
| | *dev-real* | 10.6 | $5.0_{-52.8\%}$ | $4.4_{-58.5\%}$ | $4.5_{-57.5\%}$ | $4.2_{-60.4\%}$ | $\mathbf{4.1}_{-61.3\%}$ |
| | *dev-simu* | 12.4 | $6.8_{-45.2\%}$ | $6.1_{-50.8\%}$ | $6.0_{-51.6\%}$ | $6.1_{-50.8\%}$ | $\mathbf{5.8}_{-53.2\%}$ |
| | *avg.* | 12.8 | $6.9_{-46.1\%}$ | $6.3_{-50.8\%}$ | $6.3_{-50.8\%}$ | $6.1_{-52.3\%}$ | $\mathbf{5.9}_{-53.9\%}$ |
| VB-DEMAND | *baby-cry* | 8.0 | $7.0_{-12.5\%}$ | $6.4_{-20.0\%}$ | $6.4_{-20.0\%}$ | $6.2_{-22.5\%}$ | $\mathbf{6.0}_{-25.0\%}$ |
| | *helicopter* | 8.4 | $7.4_{-11.9\%}$ | $7.1_{-15.5\%}$ | $7.2_{-14.3\%}$ | $6.9_{-17.9\%}$ | $\mathbf{6.9}_{-17.9\%}$ |
| | *crowd-party* | 22.6 | $21.4_{-5.3\%}$ | $19.9_{-11.9\%}$ | $20.1_{-11.1\%}$ | $19.5_{-13.7\%}$ | $\mathbf{19.2}_{-15.0\%}$ |
| | *avg.* | 13.0 | $11.9_{-8.5\%}$ | $11.1_{-14.6\%}$ | $11.2_{-13.8\%}$ | $10.8_{-16.9\%}$ | $\mathbf{10.7}_{-17.7\%}$ |

generation ability. Among 7b models, LLaMA-2-7b outperforms LLaMA-7b and Falcon-7b thanks to larger-scale training data and longer context length.

## D.2    Results on Different SNRs

Table 11 reports more results on different-SNR testing conditions. Similar to Table 2, we can observe consistent performance gains of RobustGER over vanilla GER and audio denosing baselines under different noise levels, i.e., ranging from 0 dB (quite noisy) to 20 dB (relatively clean). In addition, RobustGER also surpasses the reranking upper-bound $o_{nb}$ under some testing scenarios, indicating the effectiveness of RobustGER over conventional LM rescoring methods.

Furthermore, we also report error correction results on clean test data from VB-DEMAND and LS-FreeSound datasets, where significant GER improvement with 46.2% and 30.0% relative WER reductions are achieved by RobustGER approach. This experimental evidence demonstrates the excellent generality of RobustGER for various ASR scenarios.

## D.3    Ablation Study of Language Embedding Extractor

Table 13 illustrates the ablation study of proposed language-space noise embedding with different text embedding extractors. First, we try the input word-to-embedding layer in LLaMA-2-7b to extract both utterance- and token-level embeddings in §4.2, which leads to minor gains over audio denosing baseline, indicating that the LLaMA embedding is less discriminative for audio noise modeling. The supervised text classifier FastText (Grave et al., 2018) provides a better solution to extract text embeddings for modeling the N-best list diversity. Benefiting from the powerful global context modeling ability of Transformer (Vaswani et al., 2017), SBERT (Reimers & Gurevych, 2019) presents the best performance for language-space noise embedding extraction, which well represents both utterance- and token-level embeddings as shown in Table 3.

Table 15: N-best hypotheses from a speech sample under different noise conditions. We use two noise types (i.e., Babble and AC/Vacuum) and two SNR levels (i.e., 0 and 10 dB) from LibriSpeech-FreeSound test set, where the original sample id is "237-134500-0040". The "Acoustic Score" denotes the decoding score from Whisper Large-V2 model, which is calculated by negative entropy. Red font highlights the wrong tokens compared to ground-truth transcription.

| Noise Type | SNR (dB) | N-best Hypotheses | Acoustic Score | WER (%) |
|---|---|---|---|---|
| Babble | 0 | i pray for them but that is not the same as i pray for sam | −0.467 | 33.3 |
| | | i pray for them but that is not the same as i pray for science | −0.485 | 33.3 |
| | | i pray for them but that is not the same as if i prayed for sam | −0.516 | 26.7 |
| | | i pray for them but that is not the same as i pray for sons | −0.517 | 33.3 |
| | | i pray for them but that is not the same as if i pray for sam | −0.521 | 33.3 |
| | 10 | i pray for you but that is not the same as if you prayed yourself | −0.328 | 0.0 |
| | | i pray for you but that is not the same as if you prayed yourself | −0.328 | 0.0 |
| | | i pray for you but that is not the same as if you pray yourself | −0.340 | 6.7 |
| | | i pray for you but that is not the same as if you pray for yourself | −0.426 | 13.3 |
| | | i pray for you but that is not the same as if you prayed for yourself | −0.449 | 6.7 |
| AC | 0 | i pray for you but that is not the same as if you prayed yourself | −0.329 | 0.0 |
| | | i pray for you but that is not the same as if you pray yourself | −0.369 | 6.7 |
| | | i pray for you but that is not the same as if you pray for yourself | −0.388 | 13.3 |
| | | i would pray for you but that is not the same as if you prayed yourself | −0.428 | 6.7 |
| | | i pray for you but that is not the same as if you prayed for yourself | −0.429 | 6.7 |
| | 10 | i pray for you but that is not the same as if you prayed yourself | −0.305 | 0.0 |
| | | i pray for you but that is not the same as if you prayed yourself | −0.305 | 0.0 |
| | | i prayed for you but that is not the same as if you prayed yourself | −0.343 | 6.7 |
| | | i prayed for you but that is not the same as if you prayed yourself | −0.343 | 6.7 |
| | | i prayed for you but that is not the same as if you prayed yourself | −0.343 | 6.7 |
| Clean | ∞ | i pray for you but that is not the same as if you prayed yourself | −0.280 | 0.0 |
| | | i pray for you but that is not the same as if you prayed yourself | −0.280 | 0.0 |
| | | i pray for you but that is not the same as if you prayed yourself | −0.280 | 0.0 |
| | | i pray for you but that is not the same as if you prayed yourself | −0.280 | 0.0 |
| | | i pray for you but that is not the same as if you prayed yourself | −0.280 | 0.0 |
| Ground Truth | | i pray for you but that is not the same as if you prayed yourself | - | - |

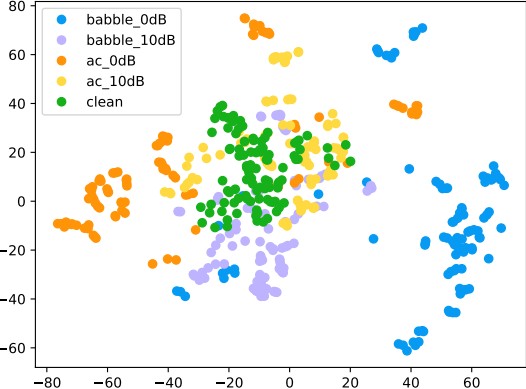

Figure 6: The t-SNE visualizations of language-space noise embeddings from source speech under different noise types and SNR levels. The average distances between embeddings of clean and various noisy conditions are: **58.6** (babble_0dB), **24.5** (babble_10dB), **22.6** (ac_0dB) and **14.3** (ac_10dB).

## D.4 ABLATION STUDY OF AUDIO NOISE DISTILLATION

Table 14 explores different KD approaches for audio noise distillation. The first one is teacher-student learning, which implements distillation by performing KL-divergence regularization between a trainable student and a frozen teacher, but it shows minor gains of performance. In comparison, contrastive learning technique achieves better results by introducing positive vs. negative samples to learn distinctiveness. However, it is still sub-optimal due to the large distance between language

and audio spaces, i.e., the anchor (language noise embedding) is far away from the positive (noisy audio embedding) and negative (clean audio embedding) samples that are relatively closer to each other. To this end, our utilized MINE introduces a neural network to estimate and maximize mutual information, which is more direct and effective in manipulating representations in different spaces for knowledge distillation. As a result, MINE achieves the best performance of audio noise distillation.

### D.5 RELATIONSHIP BETWEEN NOISY SPEECH AND N-BEST LIST DIVERSITY

As introduced in §1, our insight of proposing language-space noise embedding to represent audio noise is the relationship between the noise conditions of source speech and the diversity of decoded N-best list from ASR model, i.e., the worse noisy conditions (more challenging noise type or lower SNR), the higher uncertainty of ASR beam search decoding, and thus results in more diverse N-best hypotheses. To verify the reliability of this insight, we present the N-best hypotheses from a speech sample under different noise conditions in Table 15. For Babble noise, we can observe that 0 dB yields higher decoding uncertainty (i.e., lower acoustic scores) than 10 dB, which results in more diverse N-best hypotheses and worse 1-best WER, i.e., more language noise. Similar phenomenon can be observed in AC noise condition. On the other hand, we notice from Table 11 that Babble noise under same SNR level yields worse ASR results than AC noise, which means Babble is a more challenging noise type. As a result, Babble_0dB produces more diverse N-best list than AC_0dB, which is same for Babble_10dB and AC_10dB. In particular, the highly intelligible clean speech yields no N-best diversity. Fig. 6 visualize the language noise that originates from different audio noise, where the distances between clusters well represent the noise levels of source speech.

In summary, the relationship between the audio noise in source speech and the language noise in decoded N-best list inspires us to propose *language-space denoising*. Fortunately, the powerful generation ability of LLMs promotes the success of this research idea.

## E LIMITATIONS

Though effective in improving noisy ASR performance, there still exist some limitations in the proposed RobustGER.

- Table 16 presents a failure case on CHiME-4 *dev-real* set. There is one error in N-best hypotheses, i.e., the word "Miss" that should be "Ms" in ground truth. The GER baseline successfully corrects this error while RobustGER fails. The reason could be, the words "Ms" (/mɪz/) and "Miss" (/mɪs/) sound similar especially under noisy scenarios, GER cannot distinguish them so it depends on LLMs to decide based on context. Thanks to the rich linguistic knowledge and powerful reasoning ability, LLMs enable GER to generate the correct word "Ms" that is more appropriate than "Miss" in this context. On the other hand, with the proposed language-space denoising, RobustGER successfully perceives the trivial difference between their pronunciations but find the word is more likely to be "Miss" (*e.g.*, maybe the speaker's pronunciation is not standard). Such information misleads LLMs to generate the wrong word. Therefore, this is a problem of trade-off between contextual information and denoising for LLMs to generate correct transcription: 1) when both homophones suit the context, LLMs should be carefully in denoising to find the correct word (see Table 5), 2) when one of homophones is obviously more suitable to the context than another one, LLMs may not need denoising as it could provide misleading information. We believe this could be a promising research direction for future work on GER.

- We observe from main results in Table 1 that both GER and our RobustGER achieves significantly more improvements on CHiME-4 dataset than other datasets. This phenomenon has been also observed and analyzed in the original GER benchmark (Chen et al., 2023b), as there are many financial terminologies in the transcriptions of CHiME-4 that are relatively easier for LLMs to correct. Therefore, in future we may need a analysis of error types for CHiME-4 to understand how RobustGER works there.

- After our initial draft was released on OpenReview in October 2023, we also learned that there have been recent developments in post-recognition text modeling, as well as LLM based efforts in audio understanding (Gong et al., 2023a;b; Wu et al., 2023b) and speaker diarization (Park et al., 2023; Wang et al., 2024). We hope to align the efforts of different

Table 16: Failure case of RobustGER. The test sample is from CHiME-4 *dev-real* dataset with ID as "M03_052C010R_BUS".

| Method | Utterance | WER (%) |
|---|---|---|
| N-best Candidates | miss amsterdam declined to comment | 20.0 |
| | miss amsterdam declined to comment | 20.0 |
| | ms amsterdam declined to comment | 0.0 |
| | miss amsterdam declined to comment | 20.0 |
| | miss amsterdam decline to comment | 40.0 |
| GER | ms amsterdam declined to comment | **0.0** |
| RobustGER | miss amsterdam declined to comment | 20.0 |
| Ground Truth | ms amsterdam declined to comment | - |

Table 17: Distances between the language noise embeddings from clean and different noisy conditions. The corresponding t-SNE visualizations are presented in Fig. 4.

| Clean vs. | ac | babble | cafe | car | metro | traffic | avg. |
|---|---|---|---|---|---|---|---|
| Language Noise Emb. | **59.7** | 54.9 | 32.4 | 12.7 | 19.1 | 17.4 | 32.7 |
| + Audio Distillation | 57.6 | **87.5** | **53.2** | **37.5** | **32.1** | **51.8** | **53.3** |

research groups to enable more robust and resilient text modeling evaluations for various speech and audio processing tasks in the future, as part of a collaborative community effort.

