# OpenReview forum: "Large Language Models are Efficient Learners of Noise-Robust Speech Recognition"
_ICLR.cc/2024/Conference — ICLR 2024 spotlight_

### Official Review · Reviewer_nSVq · 2023-10-18

**Soundness:** 3 good
**Presentation:** 4 excellent
**Contribution:** 4 excellent
**Rating:** 8
**Confidence:** 4

**Summary:**

In a previous study, generative error correction (GER) is achieved by learning the mapping from ASR N-best hypotheses to ground-truth transcription through efficient LLM finetuning. This paper extends this idea and focuses on noisy conditions. To avoid the cross-modality gap, the authors propose a novel idea to extract a language-space noise embedding from the N-best list to represent the noise conditions of source speech. Furthermore, in order to enhance its representation ability of audio noise, a knowledge distillation (KD) approach via mutual information estimation (MINE) is employed. The experiments show that the proposed method can significantly outperform the conventional LM rescoring baseline. Several additional experiments are also included in the Appendix which provide more insight into the proposed RobustGER. Overall, the paper is very clear and well-written. It describes the problem and explains the solution well. The experiments done are reflective of the proposed model's performance.

**Strengths:**

1)	Novel idea to apply LLM for noise-robust ASR.
2)	Extract language-space noise embedding with knowledge distillation based on mutual information.
3)	Good performance improvement.
4)	Plenty of experiments and ablation studies.
5)	Insightful discussions, such as t-SNE visualization, and the relationship between noisy speech and n-best list diversity.

**Weaknesses:**

The way to extract the audio noise embedding is from the ASR encoder (i.e., Whisper Large-V2). This may only make sense for the Whisper ASR, as a recent paper [r1] pointed out that the noise-robustness of Whisper does not come from noise-invariant, but recognizes speech conditioned on the noise type. In summary, the Whisper encoder is a suitable model to extract noise information. On the other hand, other ASR models may not have such ability and they achieve noise-robustness through the noise-invariant encoder. If this is the case, those ASR encoders may not be suitable to extract audio noise embedding. A discussion and simple experiment about this would be great.


[r1] Gong, Y., Khurana, S., Karlinsky, L., & Glass, J. (2023). Whisper-at: Noise-robust automatic speech recognizers are also strong general audio event taggers. arXiv preprint arXiv:2307.03183.

**Questions:**

1)	As pointed out on page 5, the noise embedding is calculated by their diversity “similar to variance”, however in eq (4) and (6), the sentence embedding differences are simply summed, so I guess an abs or square operation is needed?
2)	Following the previous question, in the appendix page 16, you mentioned that the dimension of language-space noise embedding E_LN is N(N-1)xD_sbert. Could you explain where N(N-1) comes from? I cannot see this dimension from eq (4) and (6).
3)	In figure 2, and eq (3), why the language-space noise embedding is ‘subtracted’ from the prompt? I found another related equation in eq (13) of the Appendix and the authors only mention “the subtraction operation denotes “denoise””, more explanation is needed.
4)	In eq(8), IΘ(X; Y ) should be IΘ(X; Z)

---

> ### Author Response · Authors · 2023-11-15
> **Response to Reviewer nSVq**
>
> We appreciate Reviewer nSVq for considering our work is clear, novel, and solid and very thank your reviewing efforts and advice. Your comments for improvement are professional and constructive. Please find the responses below:
>
> - ***Q1: Whisper encoder is a suitable model to extract noise information. On the other hand, other ASR models may not have such ability. A discussion and simple experiment about this would be great.***
>
>   Thanks for your comment. We agree with you that Whisper encoder is better than other encoders for audio noise embedding extraction. We have added an experiment on CHiME-4 in Table 17 (Appendix), which compares Whisper encoder with WavLM encoder and Wav2vec2 encoder. Results on audio-space denoising shows that Whisper encoder performs better than WavLM and Wav2vec2, which verifies the claim above. However, since audio denoising itself does not perform well in our experiments, we may need more studies to compare these ASR encoders for noise information extraction, which we would like to leave for future work. Thanks again for bringing up this topic.
>
> - ***Q2: As pointed out on page 5, the noise embedding is calculated by their diversity “similar to variance”, however in eq (4) and (6), the sentence embedding differences are simply summed, so I guess an abs or square operation is needed ? In the appendix page 16, you mentioned that the dimension of language-space noise embedding E_LN is N(N-1)xD_sbert. Could you explain where N(N-1) comes from? I cannot see this dimension from eq (4) and (6).***
>
>   Thanks for pointing out this. We apologize for making you confused, and we have removed the term “similar to variance”.
>
>   As for Eq.(4) and (6), we also make a mistake (sorry again), please kindly refer to our updated manuscript, it should be the concatenation of all sentence embedding differences, instead of summing them up. From N-best hypotheses, we can collect $N\cdot(N-1)/2$ sentence-level embedding differences from each pair of hypotheses in the N-best list (i.e., $1 + 2 + \cdots + (N-1) = N\cdot(N-1)/2$), where each of them has tensor shape $ \mathbb{R}^{1\times D_\text{sbert}}$. Samely, we can also obtain $N\cdot(N-1)/2$ token-level embedding differences. Therefore, we concatenate all of these $N\cdot(N-1)$ embedding differences along the first dimension to get a language-space noise embedding $E_\text{LN} \in \mathbb{R}^{N\cdot(N-1)\times D_\text{sbert}}$.
>
>   We have made these clear in the updated manuscript.
>
> - ***Q3: In figure 2, and eq (3), why the language-space noise embedding is ‘subtracted’ from the prompt? I found another related equation in eq (13) of the Appendix and the authors only mention “the subtraction operation denotes “denoise””, more explanation is needed.***
>
>   Thanks for pointing out this, and we apologize for making you confused. After calculating language-space noise embedding $E_\text{LN}$, we add a minus sign before it to build a **denoise** embedding, which is then sent into LLM to teach it to perform language-space denoising (i.e., **reduce** the noise from LLM finetuning). This is an intuitive design to implement the idea of language-space **denoising**. We have made this clear in Section 4.1 and modified the “subtract” in Fig.2 to “denoise”.
>
> - ***Q4: In eq(8), $I_\Theta(X; Y)$ should be $I_\Theta(X; Z)$.***
>
>   Thanks for pointing out this, we have corrected it in the updated manuscript.

---

> > ### Comment · Reviewer_nSVq · 2023-11-20
> > **Reponse to the authors**
> >
> > Thank you for your reply!  About the subtraction operation, is it only to align the concept of **de**noise? Just want to make sure, if we remove the minus sign, the performance will still remain?

---

> ### Author Response · Authors · 2023-11-20
> **Response to Reviewer nSVq**
>
> Thank you for comment. The design of subtraction operation is **mainly** to align the concept of denoise, but also considering the performance. If we remove the minus sign, the proposed approach still outperforms the baselines but are worse than our current result with this minus sign. We speculate the reason could be, the LLaMA-Adapter itself can learn this minus sign to "denoise" instead of "add noise" during finetuning, but if we explicitly define the minus sign, it might be easier for LLaMA-Adapter to conduct denoising.
>
> Thank you again for bringing up this question, we believe it could become a promising interpretability topic for future work.

---

> > ### Comment · Reviewer_nSVq · 2023-11-22
> > **Reponse to the authors**
> >
> > Thank you for the reply!  I do not have further questions.

---

### Official Review · Reviewer_QKKW · 2023-10-28

**Soundness:** 2 fair
**Presentation:** 3 good
**Contribution:** 3 good
**Rating:** 6
**Confidence:** 4

**Summary:**

This work extends an established benchmark of generative error correction with a new "HyPoradise" dataset, in order to enable LLMs to perform error correction. The study presents application on noisy-robust speech recognition and claims that it reaches up to 53.9% improvement on word error rate.

The study itself follows the latest trend of research, where LLM is used to

**Strengths:**

1. The work itself holds certain level of novelty, with good review on earlier literatures on both error correction in ASR and LLM.
2. The methodology is clearly presented, along with the novelty of the paper.
3. With some ambiguities in the middle, the paper itself clarifies the idea with experiments subtlely.

**Weaknesses:**

1. I think the topic of error correction might be a poor fit to the conference. But perhaps I am wrong on this so correct me if so.
2. There lacks the practical discussion on additional workload, especially on resources.
3. The description of building the embedding space is somehow confusing in particular terms. For example, in Section 4.2.1 - what is "diversity similar to variance"?

Minor issues:
1. Section 5.4 - What is Table 14?
2. I suggest to put the definition of embedding a bit earlier from the beginning of Section 4. Otherwise, Figure 2 looks a bit confusing.

**Questions:**

1. I wonder the motivation of using Robust Hyporadise dataset for noisy ASR condition. What kind of noise it exactly contains? Is it replacible with other noisy datasets that are more commonly known to the ASR community, such as Switchboard and VoxCeleb (just two examples, they may not be good fit)?
2. Do you think your model will be sensitive to sampling frequency? I mentioned Switchboard in the last question, which is an 8KHz dataset.
3. In section 4.2.2, why you think MINE can enhance the noise representation ability? It looks like MINE is not part of novelty here, so any work backing it up?

**Details Of Ethics Concerns:**

This paper has no ethical concern from my point of view.

---

> ### Author Response · Authors · 2023-11-15
> **Response to Reviewer QKKW (Q1)**
>
> We appreciate Reviewer QKKW for considering our work is clearly presented with good reviews and novelty, and your comments  are very helpful for us on improving presentation quality and clarifications with wider impacts.
> Please find the corresponding responses below:
>
> - ***Q1: I think the topic of error correction might be a poor fit to the conference. But perhaps I am wrong on this so correct me if so.***
>
>   Thanks for your comment. We respectfully hold the view that ASR error correction should be a good fit to ICLR conference because:
>
>   1) Speech and ASR has been a subject area of ICLR conference [1-4], ASR error correction is also popular in similar machine learning and AI conferences recently [5-8] (including our baseline – Generative Error Correction in [5]). This shows that ASR error correction has started to become popular in the machine learning community, especially with the latest advances of LLMs.
>   2) Our ASR error correction work follows the latest research trend of adapting LLMs to various downstream tasks [9-11]. We first propose some general findings about LLMs in multimodal tasks: language embedding outperforms audio embedding in prompting LLM adaptation. Then, we propose a novel representation learning method to extract such language embedding to benefit LLM adaptation, and our experiments demonstrate its effectiveness. Therefore, our approach could be potentially applied to other multimodal tasks with LLMs in future work. We believe such contributions are also a good fit to ICLR conference.
>   3) ASR error correction itself can be viewed as a seq2seq representation learning task [5-8]: the ASR-recognized text is sent into an encoder-decoder (e.g., Transformer) or decoder-only (e.g., LLaMA) architecture to generate a corrected text as output, where the key is to learn the contextual semantics of input text, i.e., text representation learning.
>
> **Reference**
>
> [1] Chang, Oscar, Dongseong Hwang, and Olivier Siohan. "Revisiting the Entropy Semiring for Neural Speech Recognition." ICLR 2023.
>
> [2] Ren, Yi, Chen Zhang, and Y. A. N. Shuicheng. "Bag of Tricks for Unsupervised Text-to-Speech." ICLR 2023.
>
> [3] Shim, Kyuhong, Jungwook Choi, and Wonyong Sung. "Understanding the role of self attention for efficient speech recognition." ICLR 2022.
>
> [4] Shi, B., Hsu, W. N., Lakhotia, K., & Mohamed, A. Learning Audio-Visual Speech Representation by Masked Multimodal Cluster Prediction. ICLR 2022.
>
> [5] Chen, Chen, Yuchen Hu, Chao-Han Huck Yang, Sabato Macro Siniscalchi, Pin-Yu Chen, and Eng Siong Chng. "Hyporadise: An open baseline for generative speech recognition with large language models." NeurIPS 2023.
>
> [6] Leng, Yichong, Xu Tan, Linchen Zhu, Jin Xu, Renqian Luo, Linquan Liu, Tao Qin, Xiangyang Li, Edward Lin, and Tie-Yan Liu. "Fastcorrect: Fast error correction with edit alignment for automatic speech recognition." NeurIPS 2021.
>
> [7] Leng, Yichong, Xu Tan, Wenjie Liu, Kaitao Song, Rui Wang, Xiang-Yang Li, Tao Qin, Ed Lin, and Tie-Yan Liu. "Softcorrect: Error correction with soft detection for automatic speech recognition." AAAI 2023.
>
> [8] Zhao, Zewei, and Houfeng Wang. "Maskgec: Improving neural grammatical error correction via dynamic masking." AAAI 2020.
>
> [9] Yu, Wenhao, Dan Iter, Shuohang Wang, Yichong Xu, Mingxuan Ju, Soumya Sanyal, Chenguang Zhu, Michael Zeng, and Meng Jiang. "Generate rather than Retrieve: Large Language Models are Strong Context Generators." ICLR 2023.
>
> [10] Kojima, Takeshi, Shixiang Shane Gu, Machel Reid, Yutaka Matsuo, and Yusuke Iwasawa. "Large language models are zero-shot reasoners." NeurIPS 2022.
>
> [11] Li, Xuechen, Florian Tramer, Percy Liang, and Tatsunori Hashimoto. "Large Language Models Can Be Strong Differentially Private Learners." ICLR 2022.

---

> ### Author Response · Authors · 2023-11-15
> **Response to Reviewer QKKW (Q2~Q6)**
>
> - ***Q2: There lacks the practical discussion on additional workload, especially on resources.***
>
>   Thanks for pointing out this. In our experiments, we use 1 NVIDIA A40 GPU (48GB) for model training, which takes 1.5 hours for CHiME-4 dataset, 2.0 hours for VB-DEMAND, 1.6 hours for NOIZEUS, 4.5 hours for LS-FreeSound, and 3.8 hours for RATS, respectively. Actually we have introduced them in Appendix C.2 (first paragraph).
>
> - ***Q3: The description of building the embedding space is somehow confusing in particular terms. For example, in Section 4.2.1 - what is "diversity similar to variance"?***
>
>   We apologize for making you confused, we have removed this term to avoid ambiguity. Eq. (4) concatenates the embedding difference between each two hypotheses in the N-best list, which measures the diversity within the N-best hypotheses.
>
> - ***Q4: Section 5.4 - What is Table 14?***
>
>   We apologize for making you confused. Table 14 indicates the ablation study of the audio distillation approach in terms of different KD techniques (i.e. teacher-student learning, contrastive learning, and our utilized MINE), where the analysis is provided in Appendix D.4. We mention it in Section 5.4 to highlight the effectiveness of proposed audio distillation approach.
>
> - ***Q5: I suggest to put the definition of embedding a bit earlier from the beginning of Section 4. Otherwise, Figure 2 looks a bit confusing.***
>
>   Thanks for your suggestion, and we apologize for making you confused. We would like to address your concerns from two aspects:
>
>   1) Actually we had tried to put the definition of embeddings in Eq. (4)-(6) at the beginning of Section 4, but that would break its logic of presentation. Because we have to first introduce the overall framework of RobustGER (Section 4.1) and then go into the specific embedding extraction (Section 4.2) and distillation approach (Section 4.3).
>   2) Fig.2 aims to depict the overall framework (left part) and the embedding extraction (right part), where the right part just simply sketches the embedding extraction approach, instead of detailed description due to space limit. For better clarity, we have added some explanations in Fig. 2’s caption to help readers understand it, as well as a hyperlink to help them find the specific definitions in Eq. (4)-(6).
>
>   We hope this makes sense, and we are also willing to discuss more if you have any further suggestions.
>
> - ***Q6: I wonder the motivation of using Robust Hyporadise dataset for noisy ASR condition. What kind of noise it exactly contains? Is it replacible with other noisy datasets that are more commonly known to the ASR community, such as Switchboard and VoxCeleb (just two examples, they may not be good fit)?***
>
>   Thanks for your comment. Our Robust HyPoradise dataset covers many common noisy ASR scenarios in real world, in order to evaluate the generality of our proposed approach. The details of their noise conditions are introduced in Appendix A.2, which include:
>   1) **CHiME-4:** bus, cafe, pedestrian area, street junction;
>   2) **VB-DEMAND:** training data contains 10 noise types (babble, cafeteria, car, kitchen, meeting, metro, restaurant, speech-shaped noise, station, traffic), test data contains 3 new noise types (baby-cry, helicopter, and crowd-party) – simulate an unseen noise testing condition that is common in real-world scenarios;
>   3) **NOIZEUS:** 8 noise types from Aurora-2 database (suburban train noise, babble, car, exhibition hall, restaurant, street, airport and train-station noise);
>   4) **LibriSpeech-FreeSound:** 6 noise types from FreeSound database (metro, car, traffic, cafe, babble and ac/vacuum);
>   5) **RATS:** radio communication speech in ultra high frequency data category that is extremely noisy and challenging for ASR.
>
>   Yes, our RobustGER can be extended to any noisy ASR datasets, and we are exactly planning to make such extensions for future work, which would then be open sourced to benefit the community.
>
>   The two datasets you mentioned are both good fit for Robust Hyporadise, we did not select them before due to their less diverse noise categories than our utilized five datasets. We will add them for future extensions, thanks again for your suggestion.

---

> ### Author Response · Authors · 2023-11-15
> **Response to Reviewer QKKW (Q7~Q8)**
>
> - ***Q7: Do you think your model will be sensitive to sampling frequency? I mentioned Switchboard in the last question, which is an 8KHz dataset.***
>
>   Thanks for your comment. We think our model is not sensitive to sampling frequency. The whole pipeline of our system consists of two stages:
>   1) use Whisper for ASR decoding to generate N-best hypotheses;
>   2) employ LLM-based RobustGER to generate ground-truth transcription from N-best hypotheses.
>
>   The sampling frequency of source speech is directly related to stage 1), where Whisper is a general ASR model that can handle input speech of different sampling rates.
>
>   We also list the sampling rates of our selected ASR datasets here for your reference:
>   1) **CHiME-4:** 16 kHz;
>   2) **VB-DEMAND:** 48 kHz;
>   3) **NOIZEUS:** 8 kHz;
>   4) **LibriSpeech-FreeSound:** 16 kHz;
>   5) **RATS:** 8 kHz.
>
>   Our approach has achieved improvements on all of them.
>
> - ***Q8: In section 4.2.2, why you think MINE can enhance the noise representation ability? It looks like MINE is not part of novelty here, so any work backing it up.***
>
>   Thanks for your comment. MINE stands for “mutual information neural estimation” [12], which employs neural network to estimate mutual information. In this work, we leverage MINE to enhance the noise representation ability of proposed language embedding in two stages:
>   1) MINE is trained to learn accurate MI estimation;
>   2) We freeze the weights of MINE and maximize the estimated MI between our proposed language embedding and the noisy audio embedding, where we insert a tuner to make the language embedding trainable (see Fig. 3). As a result, our language embedding would be tuned to contain more information of audio noise.
>
>   This design is also similar to GAN that employs discriminator to supervise the generator. More details are presented in Appendix B.2.
>
>   Yes, MINE is followed from previous work [12], where the citation is put at two lines before Eq. (8).
>
> **Reference**
>
> [12] Mohamed Ishmael Belghazi, Aristide Baratin, Sai Rajeshwar, Sherjil Ozair, Yoshua Bengio, Aaron Courville, and Devon Hjelm. Mutual information neural estimation. In International conference on machine learning, pp. 531–540. PMLR, 2018.

---

> > ### Comment · Reviewer_QKKW · 2023-11-21
> >
> > Thanks for such fruitful comments! I do not have more questions for now ask.

---

### Official Review · Reviewer_ZWfm · 2023-10-31

**Soundness:** 4 excellent
**Presentation:** 3 good
**Contribution:** 4 excellent
**Rating:** 10
**Confidence:** 4

**Summary:**

This work addresses noisy conditions by proposing a language-space noise embedding derived from ASR hypotheses to aid denoising with large language models (LLMs)-based generative error correction (GER). A knowledge distillation strategy further enhances noise representation. Tests on various LLMs show up to 53.9% improvement in word error rate with limited training data, demonstrating the effectiveness of the proposed noise embedding and denoising ability of LLMs.

**Strengths:**

From a generally purposed GER benchmark to a more focused noise-robust problem, it is a suitable extension in depth and kind of milestone using LLM for robust ASR.

**Weaknesses:**

The illustrations of (b) GER with audio-space denoising (Zhang et al., 2023b; Fathullah et al., 2023) and (c) GER with language-space denoising (ours) are a little challenging to follow. Is denoised audio directly fed to the LLM adapter, or is there something else you want to express?

We noticed that the HP database only comes from the n-best of a few models. Is it possible to introduce more diverse system outputs from various models?

According to the method in the article, Section 4.3 should be the most important part, relatively speaking. Unfortunately, the space allocated to it in the article is too cramped—too many things to fit into this small section, which is not very reader-friendly.

**Questions:**

see above

---

> ### Author Response · Authors · 2023-11-15
> **Response to Reviewer ZWfm**
>
> We sincerely appreciate Reviewer ZWfm for considering our work is novel and solid with a noise-robust focus. Your suggestions are with decent insights for us to revise parts of the submission draft. Please find the responses below:
>
> - ***Q1: The illustrations of (b) GER with audio-space denoising (Zhang et al., 2023b; Fathullah et al., 2023) and (c) GER with language-space denoising (ours) are a little challenging to follow. Is denoised audio directly fed to the LLM adapter, or is there something else you want to express?***
>
>   Thanks for pointing out this, and we apologize for making you confused.
>
>   Here are detailed explanations of system (b) and (c):
>
>   **(b) GER with audio-space denoising:** Yes, the denoised audio embedding is directly fed into llama-adapter (Zhang et al., 2023b) to guide the LLM denoising process, you may turn to Fig.5 and Section B.1.2 in Appendix for details.
>
>   **(c) GER with language-space denoising:** Same pipeline as system (b) but we replace the audio embedding with our proposed language embedding. First we extract the language-space noise embedding from N-best list, and then employ KD to distill the real audio noise information to our extracted embedding, finally it is fed into the llama-adapter for language-space denoising.
>
>   **About the two citations here:**  Zhang et al., (2023b) proposes the llama-adapter and visual prompting strategy; Fathullah et al., (2023) leverages audio signal to prompt LLM. The baseline (b) is inspired by both of them.
>
>   We have modified Fig. 1 to make it more clear, and we added a hyperlink in Fig. 1’s caption to help readers find the details in Appendix (due to space limit we may not add more explanations there). We are willing to discuss more if you have any further suggestions.
>
> - ***Q2: We noticed that the HP database only comes from the n-best of a few models. Is it possible to introduce more diverse system outputs from various models?***
>
>   Thanks for your comment. Yes, both the existing work of HP and our proposed RobustHP database can be created using any pre-trained ASR models that support beam search decoding. Actually, we also plan to extend more diverse N-best hypotheses from various ASR models (e.g., WavLM, Wav2vec2, Conformer-Transducer) for future work, which would then be open sourced to benefit the community.
>
> - ***Q3: Cramped space allocated to Section 4.3.***
>
>   Thanks for pointing out this, and we have added some details in Section 4.3 to make it more readable.
>
>   Here we would like to explain our considerations for the organization of methodology section:
>
>   1) Actually we think Section 4.2 should be the core contribution of this paper, which proposes the language-space noise embedding for LLM denoising. More importantly, this part is the most original in the whole paper.
>   2) We agree with you that Section 4.3 (audio noise distillation) is an important part of this paper, but after all, it is built based on previous work MINE [1] and aims to enhance the noise representativeness of extracted language embedding in Section 4.2.
>
>   Based on above two considerations, we prefer to assign more space (1 page) to Section 4.2 to highlight the main contribution of this work. For Section 4.3, we have made the most effort to clearly present it in the main content (also nearly 1 page’s space) and put more details in Appendix B.2. We hope this makes sense, and we are also willing to discuss more if you have any further suggestions.
>
> **Reference**
>
> [1] Mohamed Ishmael Belghazi, Aristide Baratin, Sai Rajeshwar, Sherjil Ozair, Yoshua Bengio, Aaron Courville, and Devon Hjelm. Mutual information neural estimation. In International conference on machine learning, pp. 531–540. PMLR, 2018.

---

> > ### Comment · Reviewer_ZWfm · 2023-11-15
> > **solid work**
> >
> > Thank you for your reply. I learned a lot from this paper.

---

### Official Review · Reviewer_BbAN · 2023-10-31

**Soundness:** 4 excellent
**Presentation:** 4 excellent
**Contribution:** 4 excellent
**Rating:** 8
**Confidence:** 3

**Summary:**

This paper addresses the noise-robust ASR. The method is based on the LLM-based generative error correction (GER), but contrary to the existing approaches, it extract the noise information from the N-best hypotheses of transcription languages, not directly from the audio. The proposed method significantly outperformed the existing baseline and advanced the area of noise-robust ASR.

**Strengths:**

Originality:
While it would normally be better to reduce the noise before ASR or to estimate the noise directly from the acoustic data, the idea of doing noise estimation from N-best transcription hypotheses is interesting. It is interesting that this method avoids the difficulty of cross-modal fine-tuning by doing so. It is also compelling because it seems that humans actually perform similar processing in noisy environments.

Quality:
The paper is rich in evaluation, and the improvement interval is significant compared with the existing baselines. Also, it provides theoretical hypotheses why the proposed method works better that sound convincing.

Clarity:
The description appears complete. I have not read all the details, but I get the impression that this paper is very well organized.

Significance:
The task addressed is clearly significant because it has many practical applications. The novel approach presented in this paper is also be interesting, and I think it can potentially be applied in other modalities.

**Weaknesses:**

- It was very difficult to find anything to explicitly criticize about the technical content of the paper. There may be flaws and room for improvement, but that is no longer something to do at the peer review stage of this paper.
- If I had to say something, I was concerned that the notation seemed sometimes inconsistent.

**Questions:**

- The notation seems to be mixed up. $\mathcal{P}$ may be used in the sense of probability density function in equation (2), but this is also used to mean "prompt". In the definition of KL, the probability distribution is denoted as $\mathbb{P}$.
- I don't feel the need to use too much fancy notation like tensor product $\mathbb{P}_X \otimes \mathbb{P}_Z$ and Radon-Nikodym derivative $\log \frac{d\mathbb{P}}{d\mathbb{Q}}$.
- It may be just because I am conservative but $1e^{-2}$ looks like $1/\exp(2)$.
- $\mathbb{E}_p p \log p$ looks strange in Table 15.
- $\textit{i.e.}$ should not be italicized in standard writing convention.

---

> ### Author Response · Authors · 2023-11-15
> **Response to Reviewer BbAN**
>
> We sincerely appreciate Reviewer BbAN for considering our work is rich in evaluation, original and solid.
> Your suggestion and comments for further enhancing the submission quality are very helpful. Please find the responses below:
>
> - ***Q1: Mixed Notations of $P$.***
>
>   Thanks for pointing out this. We have modified them as:
>
>   1) use $\mathcal{P}$ to denote the probability density function in Eq. (2);
>   2) modify the prompt to $\mathcal{G}$;
>   3) use $\mathbb{P}$ to denote the probability distribution in definition of KL.
>
>   Considering 1) and 3) are both probability-related variables, we prefer to both use $P$ but in different font styles. We are willing to discuss more if you have any further suggestions.
>
> - ***Q2: No need to use too much fancy notation like tensor product $\mathbb{P}_X \otimes \mathbb{P}_Z$ and Radon-Nikodym derivative $\log\frac{d\mathbb{P}}{d\mathbb{Q}}$.***
>
>   Thanks for your suggestion. We have removed them to make a plain style.
>
> - ***Q3: $1e^{-2}$ looks like $1/\exp(2)$.***
>
>   Sorry for making you confused. We have modified it to $10^{-2}$.
>
> - ***Q4: $\mathbb{E}_p plogp$ looks strange in Table 15.***
>
>   Sorry for making you confused. We have modified it to “negative entropy”.
>
> - ***Q5: $\textit{i.e.}$ should not be italicized in standard writing convention.***
>
>   Thanks for your suggestion. We have corrected all of them in the paper.

---

### Comment · Area_Chair_TPSh · 2023-11-10
**reviewer-author discussions**

Dear All,

The reviewer-author discussion period will be from Nov. 10 to Nov. 22. For reviewers, please read the authors' responses and acknowledge it, respond to them early on in the discussion, and discuss points of disagreement. Thank you!

AC

---

### Author Response · Authors · 2023-11-15
**General Response to All Reviewers**

We sincerely thank the efforts of all reviewers, for their valuable, professional, and constructive comments!

Due to limited overlap in the reviewers’ concerns, we address the issues individually under each review. Given the feedback, we have made the following changes to the paper (coloured in blue in the revised manuscript):

- We have made modifications to Fig.1 and 2 as well as their captions to describe our proposed method more clearly.
- We have added some explanations in Section 4.1 to describe our design of language-space “denoising” more clearly.
- We have corrected previous mistakes in Eq. (4) and (6) as well as added some explanations in Section 4.2.1 to describe the embedding calculation more clearly.
- We have added some explanations in Section 4.3 to describe our MINE-based knowledge distillation method more clearly.
- We have corrected the typos and presentation issues, e.g., the mixed notations.
- We have added a simple experiment in Table 17 (Appendix) to compare different ASR encoders’ embeddings for audio-space denoising.

We thank the reviewers again and look forward to any further suggestions or discussion.

---

### Meta-Review · Area_Chair_TPSh · 2023-12-06

**Metareview:**

This paper proposes a novel method for noise-robust automatic speech recognition (ASR) based on large language models (LLMs) and generative error correction (GER). The main idea is to extract a language-space noise embedding from the N-best hypotheses of ASR and use it to guide the LLM to perform denoising. The paper also introduces a knowledge distillation technique based on mutual information estimation to enhance the noise representation of the language embedding. The paper evaluates the proposed method on various noisy ASR datasets and shows significant improvement over the baseline methods.

In terms of originality, the unconventional approach of conducting noise estimation from N-best transcription hypotheses stands out. Typically, the conventional method involves noise reduction before ASR or directly estimating noise from acoustic data. However, the innovative aspect of deriving noise estimation from N-best transcription hypotheses is intriguing. This approach circumvents the cross-modality challenges associated with cross-modal fine-tuning LLM.

The paper is well-written, clear, and well-organized. The problem and the solution are well-motivated and explained. The experiments are comprehensive and convincing. The paper also provides insightful analysis and discussion on the proposed method and its limitations. The paper makes a solid contribution to the field of noise-robust ASR and LLM adaptation.


Weaknesses

One weakness is that although the paper has showed the proposed method is clearly winning over baseline, LM_{rank}, GER by large margin on several tasks, there is a question about whether the baseline is a strong baseline. Therefore, it will be great if the authors can list the SOTA results for each task.

Moreover, as the authors stated that the proposed method doesn't perform signicantly good in very noisy conditions where the baseline WER is usually high. For example, in Table 1, in the babble condition of LS-FreeSound, the relative improvement is only 2.8% when the baseline WER is 32%. Therefore, a natural question is how the proposed method perform on Chime-5 or 6 tasks which have much higher WERs.

**Justification For Why Not Higher Score:**

Although this paper overall quality is very good, it doesn't report SOTA results on those tasks. Therefore, we don't know whether the improvement can maintain when baseline is strong.

**Justification For Why Not Lower Score:**

The paper has lots of significant contributions, espeically it is very novel in conducting noise estimation from the N-best transcription. The results are significant better than the several baseline models across all the evaluated tasks.

---

### Decision · Program_Chairs · 2024-01-16

Accept (spotlight)